# Inhibition of vascular calcification by inositol phosphates derivatized with ethylene glycol oligomers

Antonia E. Schantl[1,8], Anja Verhulst[2,8], Ellen Neven[2], Geert J. Behets [2], Patrick C. D'Haese[2], Marc Maillard [3], David Mordasini[3], Olivier Phan[3], Michel Burnier[3], Dany Spaggiari[4], Laurent A. Decosterd [4], Mark G. MacAskill [5], Carlos J. Alcaide-Corral[5], Adriana A.S. Tavares[5], David E. Newby[5], Victoria C. Beindl[1], Roberto Maj[6], Anne Labarre[7], Chrismita Hegde[7], Bastien Castagner[7], Mattias E. Ivarsson[6,9]* & Jean-Christophe Leroux [1,9]*

*Myo*-inositol hexakisphosphate (IP6) is a natural product known to inhibit vascular calcification (VC), but with limited potency and low plasma exposure following bolus administration. Here we report the design of a series of inositol phosphate analogs as crystallization inhibitors, among which 4,6-di-*O*-(methoxy-diethyleneglycol)-*myo*-inositol-1,2,3,5-tetrakis (phosphate), $(OEG_2)_2$-IP4, displays increased in vitro activity, as well as more favorable pharmacokinetic and safety profiles than IP6 after subcutaneous injection. $(OEG_2)_2$-IP4 potently stabilizes calciprotein particle (CPP) growth, consistently demonstrates low micromolar activity in different in vitro models of VC (i.e., human serum, primary cell cultures, and tissue explants), and largely abolishes the development of VC in rodent models, while not causing toxicity related to serum calcium chelation. The data suggest a mechanism of action independent of the etiology of VC, whereby $(OEG_2)_2$-IP4 disrupts the nucleation and growth of pathological calcification.

[1] Institute of Pharmaceutical Sciences, Department of Chemistry and Applied Biosciences, ETH Zurich, Zurich, Switzerland. [2] Laboratory of Pathophysiology, University of Antwerp, Antwerp, Belgium. [3] Service of Nephrology and Hypertension, Lausanne University Hospital, Lausanne, Switzerland. [4] Division of Clinical Pharmacology, Lausanne University Hospital, Lausanne, Switzerland. [5] University-BHF Centre for Cardiovascular Science, Queen's Medical Research Institute, University of Edinburgh, Edinburgh, UK. [6] Inositec Inc., Zurich, Switzerland. [7] Department of Pharmacology & Therapeutics, McGill University, Montreal, Canada. [8] These authors contributed equally: Antonia E. Schantl, Anja Verhulst. [9] These authors jointly supervised this work: Mattias E. Ivarsson, Jean-Christophe Leroux. *email: mattias@inositec.com; jleroux@ethz.ch

Vascular calcification (VC) is the consequence of pathological deposition of calcium phosphate mineral in soft tissues[1]. Symptoms and associated risks vary depending on the location (artery vs. valve, tunica media vs. intima) at which calcification occurs, as well as on its nature (nano- vs. micro- vs. macrocalcification)[2,3]. Clinical consequences may include aortic stiffening[4], occlusive lesions (e.g., atherosclerotic plaques)[5], or aortic valve stenosis[6,7], ultimately resulting in an increased risk for cardiovascular morbidity and mortality[8].

In chronic kidney disease (CKD), mineral homeostasis is disturbed due to impaired renal function[9], and cardiovascular complications are the leading cause of death in these patients[10,11]. Furthermore, VC is also a hallmark of a number of orphan diseases, such as pseudoxanthoma elasticum and generalized arterial calcification of infancy[12,13]. To date, there are no approved pharmacological interventions for the direct prevention or treatment of VC. The etiology of VC is complex and multifactorial. Several factors govern the calcification process in the vessel wall microenvironment, acting to promote or inhibit calcification in conjunction with local precipitation of calcium and phosphate[8]. The most important endogenous calcification inhibitors include serum fetuin-A[14], matrix Gla protein[15], and pyrophosphate (PP$_i$)[16]. Most drug candidates currently under clinical investigation for VC (e.g., bisphosphonates, phosphate-binders, vitamin K, or magnesium) act by correcting the imbalance of calcification promoters (i.e., hyperphosphatemia) and inhibitors in VC-affected individuals. However, these drug candidates have yet to prove their efficacy in cardiovascular disease patients[17,18].

Under non-pathological conditions, surges of plasma phosphate or calcium, or local drops in the concentration of circulating calcification inhibitors are buffered by acidic serum proteins, notably fetuin-A, via the formation of colloidal calci-protein particles (CPPs)[19]. Primary CPPs (CPP1) are spherical complexes of amorphous calcium and phosphate with fetuin-A with a hydrodynamic radius ($R_h$) of less than 100 nm[20]. The nanoparticles formed prevent precipitation of calcium and phosphate (i.e., hydroxyapatite formation)[21]. In the pathology of VC, the system becomes depleted of factors delaying calcification propensity and the balance between CPP1 formation and absorption is distorted[22]. As a consequence, larger ($R_h > 100$ nm), crystalline, and spindle-shaped CPPs (secondary CPPs, CPP2s) emerge[20,23]. Primary and secondary CPPs are predominantly cleared by liver sinusoidal endothelial cells and liver Kupffer cells, respectively. Scavenger receptor A is a major endocytosis receptor for CPP2s but not for CPP1s, indicating separate clearance pathways[24]. Native CPPs in fresh plasma are smaller and lower in density than synthetic CPP1 and CPP2[29]. In CKD, CPPs are likely involved in the pathology of VC[25,26]. Serum CPP levels increase with decline of renal function and correlate with coronary artery calcification score, arterial stiffness, serum phosphate, inflammation, and procalcific factors[27,28]. Moreover, in patients with advanced CKD, the $R_h$ of CPP2 was associated with VC[29]. The fact that CPP2s have the ability to induce VSMC calcification in vitro has raised the hypothesis that CPP2s may present a valid therapeutic target in VC[30].

In patients with VC, a continuum of calcification typically occurs with evidence of nano-, micro-, and macrocalcification, which follows a common pathophysiology. Scanning electron microscopy of various diseased human aortic valve and arterial tissues reveals the presence of spherical, crystalline hydroxyapatite particles ranging in size from 100 nm to 5 μm[31], suggesting that VC may follow a common process independent of the etiology[31,32]. Further evidence for the presence of micro-calcifications comes from positron emission tomography (PET) imaging of high-risk coronary atherosclerotic plaque[33] employing [$^{18}$F] sodium fluoride ($^{18}$F-NaF), a marker of active vascular

micro-calcification[34]. Subsequently, larger confluent deposits occur, which are visible on clinical computed tomography scanning[35]. Macrocalcifications negatively impact cardiovascular tissue (e.g., arterial and valvular stiffening, intima-media thickening) and hemodynamic parameters (e.g., pulse wave velocity, aortic valve pressure gradient). As a consequence, the risk for atherosclerotic events, heart failure, and cardiovascular as well as all-cause mortality increases[35–38].

Myo-inositol hexakisphosphate (IP6) is a ubiquitously occurring natural product and endogenous intracellular co-factor in mammalian cells with putative functions in the areas of vesicle recycling and nuclear transport[39]. It inhibits the formation and growth of hydroxyapatite microcrystals in soft tissues when administered parenterally[40,41]. Completed phase II studies in hemodialysis patients with calciphylaxis or coronary artery calcifications, respectively, showed good safety, tolerability, as well as encouraging efficacy[42–44]. IP6 is injected intravenously through the dialysis machine during hemodialysis sessions to allow it to reach therapeutic exposure despite its modest potency and short plasma half-life[45]. Both factors complicate its development as a chronic ambulatory therapy[46]. Given the paucity of data on the properties of IP6 analogs in this context, this study investigates whether the chemical modification of IP6 could produce more potent inhibitors with improved pharmacokinetic (PK) properties that may enable its administration by a more convenient route. In previous work from our group, where calcium phosphate nanoparticles for gene delivery were developed, it was shown that coating of these particles with conjugates of oligo(ethylene glycol) (OEG) and myo-inositol pentakisphosphate prevent the rapid size increase of calcium phosphate nanoparticles[47,48]. In the present study, structurally related conjugates are tested in the context of VC inhibition[49]. Several OEGylated myo-inositol phosphate derivatives are synthesized and screened for their ability to inhibit calcification, as well as for their resistance towards hydrolysis. The lead candidate is characterized for its efficacy in primary human vascular smooth muscle cells (VSMCs) and calcified human cardiovascular tissue explants, its biopharmaceutical properties, as well as its in vivo activity in rat models of VC[50]. This work identifies a potent OEG-derivatized inositol phosphate, (OEG$_2$)$_2$-IP4, with a distinct mechanism of action that may find clinical utility for the prevention and/or treatment of VC.

## Results

**Design and synthesis of inhibitors of VC.** Initially, OEG$_{12}$-IP5, a low molecular weight derivative of the chelator previously used by our group to stabilize calcium phosphate nanoparticles for gene delivery purposes[47], was synthesized in three steps starting from 1,3,5-O-methylidyne-myo-inositol (Supplementary Information 3.1.1.). It was purified by size-exclusion chromatography and characterized by proton-1 nuclear magnetic resonance ($^1$H-NMR) and phosphorus-31 nuclear magnetic resonance ($^{31}$P-NMR) spectroscopies (Supplementary Information 5.), and electrospray ionization-mass spectrometry (ESI-MS) (Supplementary Table 1). Starting from OEG$_{12}$-IP5, other inhibitors were designed as follows: (i) increasing the number of linked OEG units while decreasing that of phosphates (e.g., (OEG$_2$)$_2$-IP4 and (OEG$_3$)$_3$-IP3) to study the trade-off between lower charge and increased steric hindrance, (ii) replacing phosphates by sulfate groups (i.e., OEG$_{11}$-IP2S3) to investigate chemical stability towards hydrolysis and to reduce chelation of soluble Ca$^{2+}$[51], and (iii) reducing OEG chain length (OEG$_{12}$, OEG$_{11}$, OEG$_7$, and OEG$_2$) to study the influence of the latter on activity (Fig. 1a). Ten compounds, covering a broad range of chemical modifications, were produced

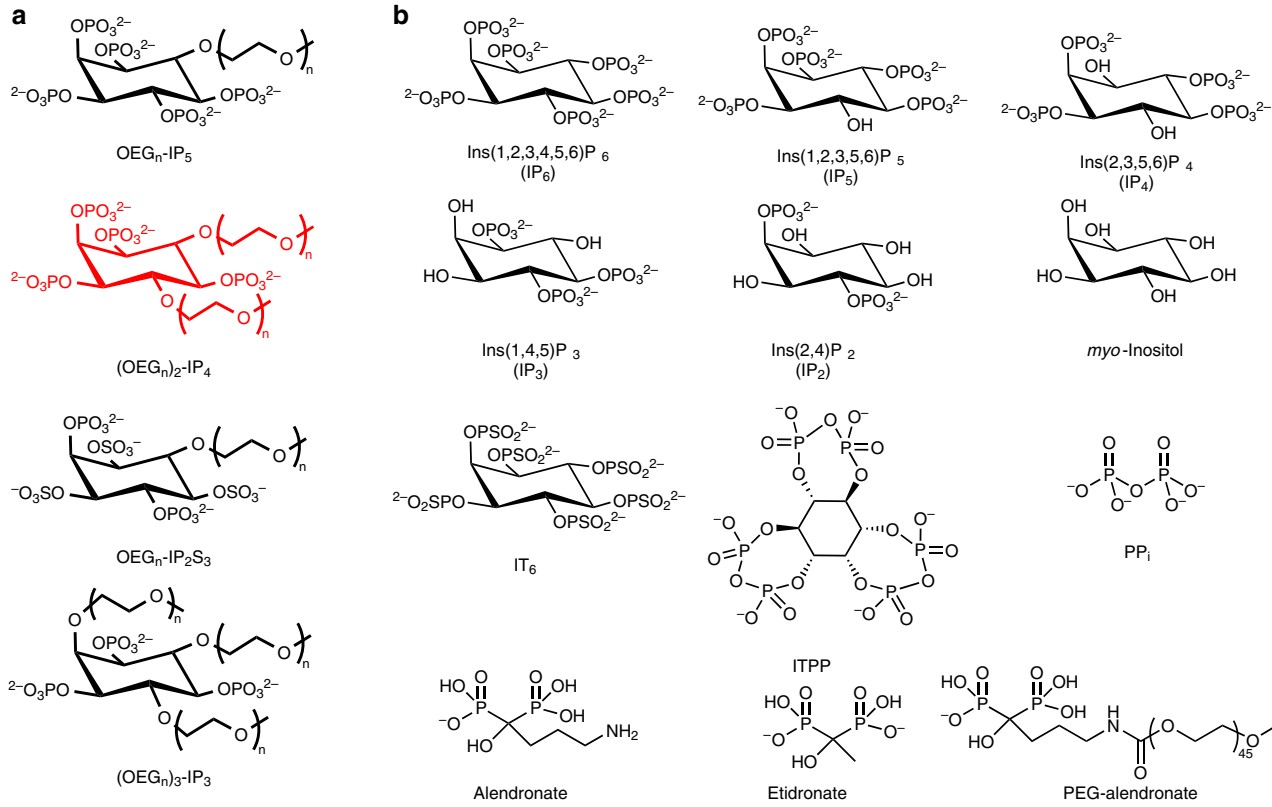

**Fig. 1 Structures of functionalized inositol phosphates and control compounds. a** Structure of the series of calcification inhibitors that were synthesized, with $n = 2, 7, 11,$ and $12$ for $OEG_n$-IP5, $n = 2, 7,$ and $11$ for $(OEG_n)_2$-IP4, $n = 11$ for $OEG_n$-IP2S3, and $n = 2$ and $7$ for $(OEG_n)_3$-IP3. Represented in red is the backbone structure of the lead inhibitor of this study. **b** Structures of compounds used as controls, including dephosphorylated (IP5, IP4, IP3, IP2, *myo*-inositol) and phosphorylated (*myo*-inositol trispyrophosphate (ITPP)) IP6 derivatives, *myo*-inositol hexathiophosphate (IT6), pyrophosphate (PP$_i$), as well as alendronate and etidronate (non-hydrolyzable analogs of PP$_i$). PEG$_{45}$-alendronate has been described previously to stabilize calcium phosphate nanoparticles in vitro[48].

or purchased, and details on their synthesis and characterization are provided in the Supplementary Methods. These compounds were compared to 12 controls (Fig. 1b).

**Drug candidate screening assay**. The compounds were ranked for calcification inhibition activity using a nanoparticle-based test, developed for risk stratification, which measures the propensity for calcification in patient serum[22]. The transformation of amorphous CPP1s to larger crystalline CPP2s in pooled human serum was monitored in the presence of inhibitor, and artificially elevated calcium and phosphate concentrations at 37 °C. The time, after which 50% of CPP phase transition occurred ($T_{50}$), was assessed by time-resolved spectrophotometry at 550 nm (Supplementary Fig. 1a). To rank the compounds' activity, a $T_{50}$ vs. compound concentration plot was generated, and the concentration necessary to delay $T_{50}$ to 350 min (c350, at which IP6's activity was at half-maximum) was derived from nonlinear curve fitting (Fig. 2a, Supplementary Figs. 1–3). $(OEG_{11})_2$-IP4 displayed an almost 10-fold greater molar activity in delaying serum calcification propensity (c350 = $3.8 \pm 0.3$ µM) than IP6 (c350 = $35.0 \pm 9.9$ µM). Further, as seen in Table 1, $(OEG_{11})_2$-IP4 was the most active molecule of all screened functionalized inositol phosphates and tested controls. However, looking at activity by unit mass, $(OEG_2)_2$-IP4 had an apparent 32% stronger inhibition than $(OEG_{11})_2$-IP4 due to its lower molecular weight—an important consideration for dosing purposes (Table 1). These data indicate that (i) replacement of two, rather than one or three phosphates of IP6 by OEG is optimal, (ii) replacement of phosphates by sulfates in the OEGylated molecules reduces activity,

and (iii) longer OEG chains increase molar activity but reduce activity by unit mass. Moreover, molar activity of $(OEG_2)_2$-IP4 was approximately six- and twofold higher than pyrophosphate (PP$_i$) (a strong endogenous inhibitor of VC) and etidronate (a hydrolysis-resistant analog of PP$_i$), respectively (Table 1, Supplementary Fig. 3).

**In vitro characterization**. To confirm the screening assay data, the hydrodynamic diameter of the CPPs was measured by dynamic light scattering (DLS). In the absence of inhibitors or in the presence of IP6 (up to 100 µM), the CPP1 size grew from roughly 150 nm after 1 h to ~1 µm after 3 h of incubation in serum (Supplementary Fig. 4c, f), indicating transition to CPP2. The CPP2 size deduced by DLS is slightly larger than the size derived from transmission electron microscopy (TEM) (Supplementary Figs. 4c and 7a), the latter being in line with previous work[52]. $(OEG_2)_2$-IP4, added to CPP1s at a final concentration of 30 µM, stabilized particle growth (Supplementary Fig. 4h)[29]. TEM revealed the presence of spherical-shaped CPP1s in samples incubated with 100 µM $(OEG_2)_2$-IP4 vs. spindle-shaped CPP2s in samples containing 100 µM IP6 (Fig. 2b). Corresponding selected area electron diffraction analysis of CPP with 100 µM IP6 showed bright spots and well-defined rings, indicating crystalline material, whereas the $(OEG_2)_2$-IP4-stabilized CPP1s were largely amorphous. The derived interplanar- or $d$-spacings for CPP incubated with IP6 were: $d1 = 0.287$ nm*, $d2 = 0.369$ nm†, and $d3 = 0.403$ nm, which are in line with reported literature $d$-spaces for hydroxyapatite: 0.28 nm*, 0.32 nm, 0.34 nm†, 0.47 nm, and

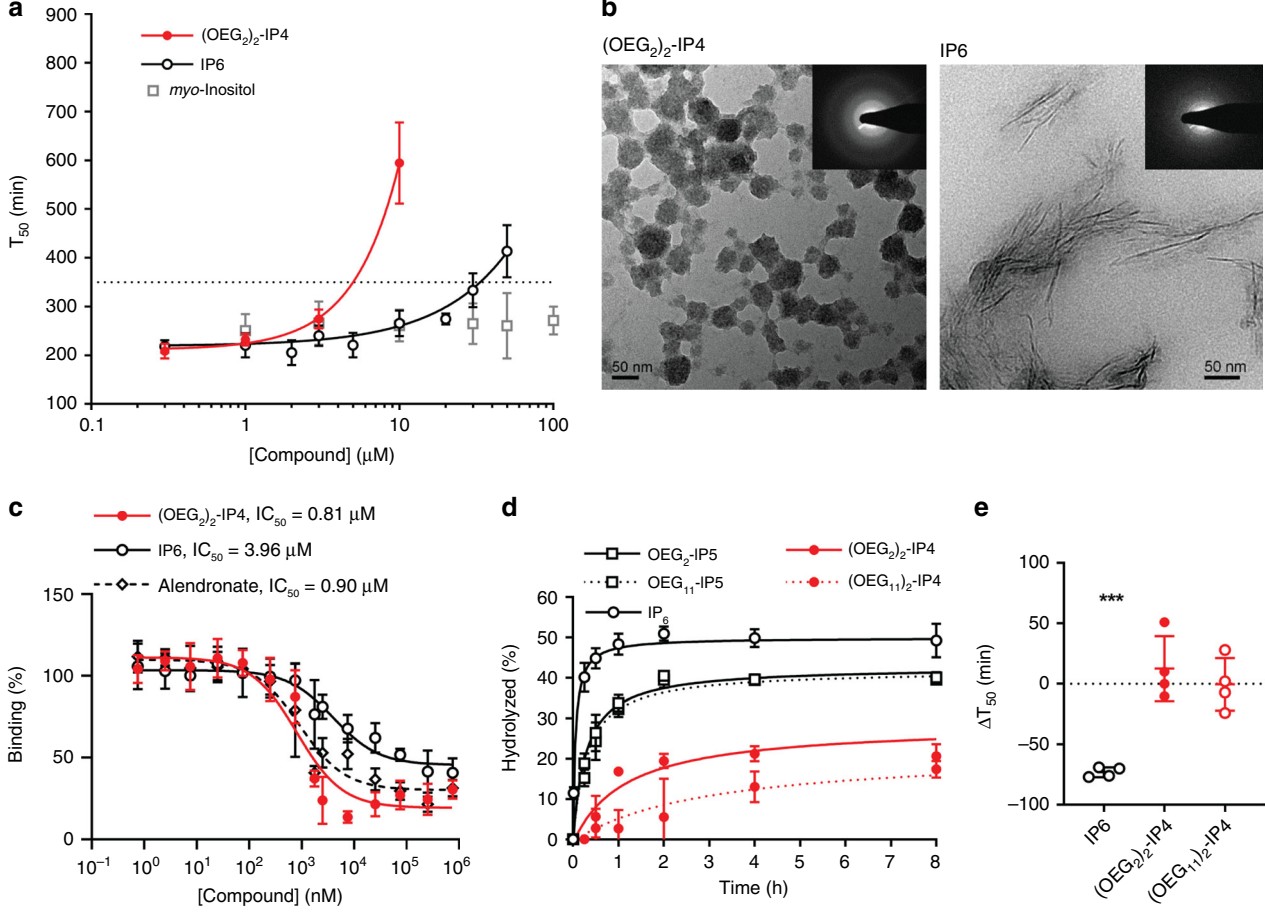

**Fig. 2 In vitro activities of functionalized inositol phosphates. a** Compounds were tested in an in vitro serum calcification propensity assay. $T_{50}$ is defined as the time at which 50% of CPP1 is transformed to CPP2. The concentration necessary to delay $T_{50}$ to 350 min (dotted line) was used to compare the compounds' activities ($n = 4$ for myo-inositol, $n = 6$ for $(OEG_2)_2$-IP4, and $n = 26$ for IP6). **b** Representative TEM images and corresponding selected area electron diffraction patterns of CPPs, showing the predominant presence of spherical-shaped CPP1s and spindle-shaped CPP2s after incubation with 100 μM $(OEG_2)_2$-IP4 and IP6, respectively for 14 h at 37 °C (scale bar = 50 nm). **c** Competitive $^{18}$F-NaF-binding assay to characterize interaction of compounds with CPPs ($n = 3$). **d** Phytase hydrolysis of compounds ($n = 3$). **e** Activity in the serum calcification propensity assay after incubation in enzymatically active human serum for 4.5 h at 37 °C ($n = 4$). All data are expressed as mean ± s.d. from $n$ independent experiments. Statistical significance was calculated from a paired Student's $t$-test, ***$p < 0.001$ vs. baseline $T_{50}$.

0.52 nm[53] (matching $d$-spaces between samples are indicated with * and †) and suggest identical material.

A $^{18}$F-NaF-based competitive binding assay was developed to indirectly assess the inhibitor affinity ($IC_{50}$) to CPPs. The uptake mechanism of $^{18}$F-NaF, a marker of micro-calcification activity by chemisorption onto hydroxyapatite crystals, is contingent on the rapid exchange of hydroxyl ($OH^-$) groups present on the hydroxyapatite surface ($Ca_5(PO_4)_3(OH)$) by $^{18}F^-$ to form fluoroapatite ($Ca_5(PO_4)_3^{18}F$)[34,54]. Similarly to $^{18}$F-NaF, IP6 and $(OEG_2)_2$-IP4 are expected to be adsorbed to the hydroxyapatite core of CPPs. $(OEG_2)_2$-IP4 reduced $^{18}F$ binding to CPP more efficiently than IP6 ($IC_{50} = 0.81$ μM, 95% CI [0.48, 1.37] vs. 3.96 μM, 95% CI [2.03, 7.75]), despite the lower negative charge density, and thus lower $Ca^{2+}$ chelating ability of $(OEG_2)_2$-IP4 compared to IP6 (Fig. 2c), hence supporting the hypothesis that the OEG moiety contributes to the activity of $(OEG_2)_2$-IP4. Alendronate showed a comparable CPP-binding affinity ($IC_{50} = 0.90$ μM, 95% CI [0.54–1.50]) while having a twofold lower activity in the calcification propensity assay ($c_{350} = 9.9 ± 1.0$ μM). Due to the electrostatic nature of the interaction of $(OEG_2)_2$-IP4 with hydroxyapatite, the binding is likely reversible. Because of these results and the fact that clinical trials using bisphosphonates in VC reported important side effects[55], alendronate was not further investigated in this study.

Since IP6 is highly susceptible to degradation to lower inositol phosphates due to endogenous phosphatase activity[39], the in vitro stability of selected OEGylated inositol phosphates after exposure to 3-phytase and incubation in enzymatically active human serum was investigated. Enzymatic hydrolysis of inositol phosphates in buffer was reduced and delayed by the presence of OEG moieties compared to IP6 (Fig. 2d), likely due to steric hindrance hampering enzymatic activity. Protection from phosphate hydrolysis increased with number, but not with length of conjugated OEG. OEGylated inositol phosphates retained complete activity in the calcification propensity assay after a 4.5-h incubation in enzymatically active serum, as opposed to IP6, with which $T_{50}$ decreased by 73 min from baseline (Fig. 2e).

All subsequent testing was performed with $(OEG_2)_2$-IP4 because it demonstrated comparable stability and molar activity to $(OEG_{11})_2$-IP4, but would eventually be administered in vivo at a lower dose in mass, given its lower molecular weight.

**Inhibition of human VSMC calcification.** A postulated driving force for VC at the cellular level are VSMCs, the main cell type of the tunica media. VSMCs can transdifferentiate towards an

**Table 1 Activities of compounds in the screening assay.**

| Compound | Concentration at $T_{50}$ = 350 min mean ± s.d. ($n$) | |
|---|---|---|
| | µM | mg/L |
| *Inositol phosphate derivatives* | | |
| OEG$_2$-IP5 | 7.9 ± 2.7 (3) | 7.1 ± 2.4 (3) |
| OEG$_7$-IP5 | 6.7 ± 0.9 (3) | 7.6 ± 1.0 (3) |
| OEG$_{11}$-IP5 | 5.3 ± 0.7 (3) | 6.8 ± 0.9 (3) |
| OEG$_{12}$-IP5 | 4.9 ± 0.6 (5) | 6.6 ± 0.9 (5) |
| (OEG$_2$)$_2$-IP4 | 4.9 ± 0.8 (6) | 4.3 ± 0.7 (6) |
| (OEG$_7$)$_2$-IP4 | 4.5 ± 0.4 (3) | 6.0 ± 0.6 (3) |
| (OEG$_{11}$)$_2$-IP4 | 3.8 ± 0.3 (3) | 6.3 ± 0.6 (3) |
| OEG$_{11}$-IP2S3 | 18.6 ± 1.9 (5) | 22.9 ± 2.3 (5) |
| (OEG$_2$)$_3$-IP3 | 7.1 ± 0.9 (3) | 5.4 ± 0.7 (3) |
| (OEG$_7$)$_3$-IP3 | 9.4 ± 0.6 (3) | 14.3 ± 1.0 (3) |
| *Inositol controls* | | |
| *myo*-Inositol | >100 (4) | >18.0 (4) |
| IP2 | 35.0 ± 7.7 (3) | 15.0 ± 3.3 (3) |
| IP3 | 18.6 ± 3.1 (3) | 10.2 ± 1.7 (3) |
| IP4 | 19.5 ± 0.7 (3) | 13.2 ± 0.5 (3) |
| IP5 | 13.2 ± 1.3 (3) | 10.5 ± 1.0 (3) |
| IT6 | 14.4 ± 0.3 (3) | 36.14 ± 0.24 (3) |
| ITPP | >100 (4) | >60.6 (4) |
| *Known inhibitors of calcification* | | |
| IP6 | 35.0 ± 9.9 (26) | 30.9 ± 8.7 (26) |
| Magnesium citrate | >100 (3) | >21.4 (3) |
| PP$_i$ | 27.4 ± 2.8 (3) | 7.3 ± 0.8 (3) |
| Etidronate | 10.5 ± 1.1 (4) | 2.6 ± 0.3 (4) |
| Alendronate | 9.9 ± 1.0 (6) | 3.2 ± 0.3 (6) |
| *Others* | | |
| PEG-alendronate | 12.3 ± 3.0 (3) | 28.1 ± 6.9 (3) |

osteoprogenitor cell type under prolonged mineral exposure or uremic stress, and release vesicles packed with high concentrations of calcium and phosphate serving as nidi for extracellular matrix calcification[56–60].

Incubating primary human VSMCs with (OEG$_2$)$_2$-IP4 (0.01–100 µM) did not lead to any changes in cell viability (Fig. 3a). The activity of (OEG$_2$)$_2$-IP4 in inhibiting calcification of VSMCs was then examined using a high-CaP (2.7 mM calcium, 2.5 mM phosphate) medium (Supplementary Fig. 5). Incubation of VSMCs in high-CaP medium induced cell monolayer calcification after 5 days (Fig. 3b). Concurrent incubation with (OEG$_2$)$_2$-IP4 (0.01–100 µM) dose-dependently prevented calcification as evidenced from the quantitative measurement of the calcium content (Fig. 3b), and qualitative staining for calcium with Alizarin Red (Fig. 3c). The effect became significant from 0.1 µM (OEG$_2$)$_2$-IP4 on, where a 54% reduction in calcification was observed compared to untreated high-CaP medium-exposed cells. Calcium deposition was almost completely abrogated above 1 µM (OEG$_2$)$_2$-IP4. IP6 reduced VSMC mineralization by ~50–70% for all doses studied (up to 100 µM). The less efficient inhibition in calcification may be due to the limited IP6 solubility in high calcium medium[51] or potential extracellular effects (e.g. protein binding)[39]. In the present model, cell viability seemed to be slightly decreased in low-dose treatment groups (0.01 and 0.1 µM (OEG$_2$)$_2$-IP4) vs. high-CaP medium control but this effect disappeared above 1 µM for (OEG$_2$)$_2$-IP4 and 10 µM for IP6 (Fig. 3d).

CPPs have been implicated in the mineralization paradox of CKD, and previous in vitro work using synthetic CPP2s suggests a role in promoting VSMC calcification[25,30,61,62] and in reducing hydroxyapatite mineral stress in macrophages[63]. Macrophage accumulation and inflammation coincide with the initial phase of calcification[3]. To this end, VSMCs and THP-1-derived

macrophages were exposed to CPP2-supplemented medium. CPP2 prepared from human serum as described previously[61] (Supplementary Figs. 6 and 7), induced a pronounced VSMC monolayer calcification after 48 h, which was decreased by (OEG$_2$)$_2$-IP4 in a dose-dependent fashion, and the effect became significant at 10 µM (Fig. 3e). These results were supported by Alizarin Red staining (Supplementary Fig. 8). (OEG$_2$)$_2$-IP4 treatment did not reduce VSMC viability in this setup (Fig. 3f). Incubating VSMCs with CPP1 increased calcium deposition vs. control, but not significantly, and (OEG$_2$)$_2$-IP4 treatment starting at 10 µM reduced calcium deposition to the level of control (Fig. 3g). Both H$_2$O$_2$ (proxy for oxidative stress) and tumor necrosis factor-alpha (TNF-α) (proxy for inflammatory response) levels were assayed in the VSMC culture supernatants subsequent to CPP2 treatment with and without compounds[30]. However, no increase in H$_2$O$_2$ nor TNF-α compared to control could be detected in either assay. For the CPP-induced VSMC calcification assay (Fig. 3e) a roughly 10-fold higher concentration of (OEG$_2$)$_2$-IP4 was necessary to achieve similar activity as in the CaP-induced VSMC calcification assay (Fig. 3b). This may be explained by a more potent induction of calcification in the CPP-induced assay, requiring higher (OEG$_2$)$_2$-IP4 concentration to counteract the procalcific environment. In the CaP assay, low inhibitor concentration may be sufficient to inhibit crystal nucleation, whereas for the CPP assay higher inhibitor concentrations may be needed to achieve sufficient particle stabilization by chemisorption of inhibitor onto CPPs. Similar results have been reported for hydrogen sulfide in the context of VC[62,64]. In both the VSMC and macrophage assays, precipitation in culture medium occurred in treatment groups containing high concentrations of IP6 (100 µM) but not with (OEG$_2$)$_2$-IP4 at the same concentrations (Fig. 3h microscopy images and Supplementary Fig. 9). This effect is likely due to the strong chelating potential of IP6 with free ionized serum calcium. Indeed, the combined treatment of 100 µM IP6 and CPP2 (i) resulted in precipitation visible to the naked eye in the treatment medium, (ii) significantly reduced macrophage viability by 24% (Fig. 3i), and (iii) TNFα (indicator for inflammatory response) release from macrophages increased 49-fold (Fig. 3h). An increase in the TNFα levels in the cell culture supernatants was not observed for (OEG$_2$)$_2$-IP4 with (Fig. 3h) and without (Supplementary Fig. 10) combined CPP2 treatment.

**Target engagement in calcified human explants**. To investigate (OEG$_2$)$_2$-IP4 interaction with calcified tissues in a clinically relevant context, a competitive binding assay using $^{18}$F-NaF was applied on severely calcified human femoral arteries and native aortic valves ex vivo. The tissue explants were incubated with (OEG$_2$)$_2$-IP4 at a concentration inhibiting 90% $^{18}$F-NaF binding (IC$_{90}$ = 8.6 µM), as assessed in Fig. 2c. First, $^{18}$F-NaF uptake was measured on tissue samples treated with vehicle (human serum) and thereafter, using the same tissue samples, treated with (OEG$_2$)$_2$-IP4 in serum. Compared to the vehicle controls, results showed a significant reduction of $^{18}$F-NaF uptake in calcified human femoral arteries (−38%) (Fig. 3j, l) as well as in calcified leaflets from native human aortic valves (−33%) (Fig. 3k, m).

**OEGylation improves the PK profile of inositol phosphates**. The plasma half-life ($t_{1/2}$) of IP6 is short (8 min) after intravenous (i.v.) bolus administration in rats[45,65], and values of 27 and 54 min after a 4-h i.v. infusion have been reported in hemodialyzed patients and healthy volunteers, respectively[43]. Therefore, the PK profile of (OEG$_2$)$_2$-IP4 was assessed in healthy and uremic rats and compared to that of IP6. At 10 mg/kg, (OEG$_2$)$_2$-IP4 displayed an almost 10-fold increase in plasmatic $t_{1/2}$ vs. IP6 after i.v. bolus

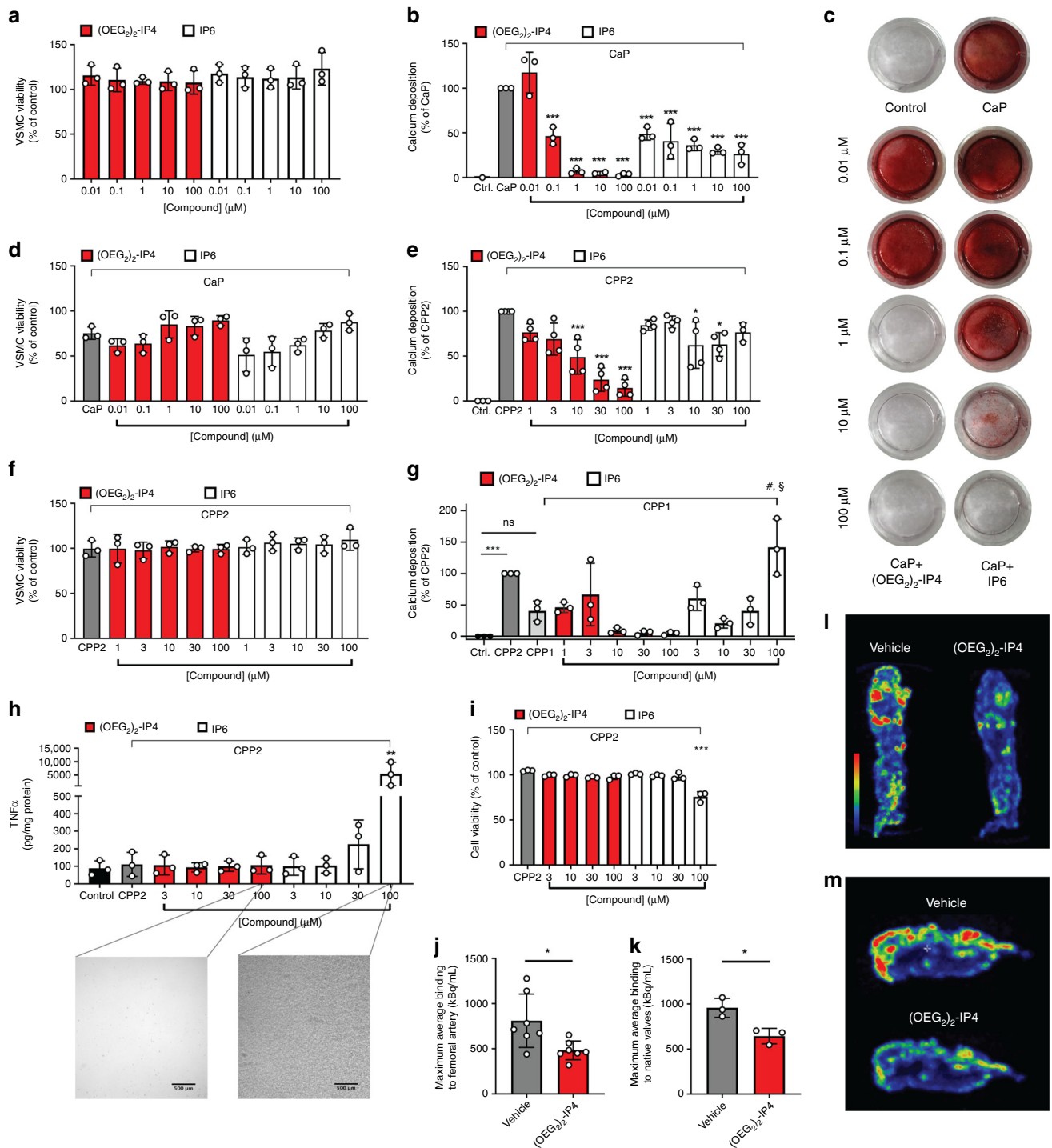

injection into male rats ($t_{1/2} = 78 \pm 32$ vs. $t_{1/2} = 8 \pm 2$ min, respectively) (Fig. 4a, b). The apparent volumes of distribution were low with $234 \pm 32$ vs. $958 \pm 405$ mL/kg for $(OEG_2)_2$-IP4 and IP6, respectively, which is generally expected for hydrophilic compounds. The maximum plasma concentration ($C_{max}$) increased fourfold, the area under the concentration vs. time curve ($AUC_{0-t}$) increased sevenfold and the plasma clearance (CL) decreased ninefold compared to IP6 (Fig. 4b). Subcutaneous (s.c.) administration of 10 mg/kg $(OEG_2)_2$-IP4 resulted in an AUC comparable to that of 100 mg/kg IP6 ($AUC_{0-360} = 1197 \pm 36$ vs. $AUC_{0-90} = 1165 \pm 76$ μM min), despite the 10-fold difference in doses (Fig. 4c, d, Supplementary Table 2)[45]. The mean residence time (MRT) was longer for $(OEG_2)_2$-IP4 compared to IP6 after s.c. ($78 \pm 5$ vs.

$45 \pm 20$ min) as well as i.v. ($30 \pm 3$ vs. $14 \pm 6$ min) administrations (Supplementary Fig. 11).

No significant accumulation occurred following repeated s.c. administration of 50 mg/kg $(OEG_2)_2$-IP4 for 7 days (Fig. 4e). Moreover, the systemic exposure of the compound after s.c. administration was assessed in rats with VC (Fig. 4f, Supplementary Table 3). In the presence of cardiovascular calcification, the plasma exposure of $(OEG_2)_2$-IP4 was significantly higher at 4 and 6 h after dosing, but the $AUC_{0-1440}$ was not statistically different to the value obtained before the induction of calcification ($AUC_{0-1440} = 9077 \pm 2455$ vs. $6049 \pm 799$ μM min).

For many drugs, the PK profile depends on the patient's renal status. Therefore, $(OEG_2)_2$-IP4 PK profile was also determined in

**Fig. 3 $(OEG_2)_2$-IP4 abrogates in vitro VSMC calcification and $^{18}$F-NaF uptake onto calcified human explants. a, d, f** VSMC viability as determined by the MTS assay compared to medium control. **b, c** VSMC calcification was induced by high-CaP medium and the effect of cell treatment with IP6 or $(OEG_2)_2$-IP4 was investigated quantitatively by colorimetric analysis of cellular calcium deposition after 5 days control (calcium deposition in mass units in Ctrl. ranged between 81 and 622 μg Ca/mg protein) (**b**) and qualitatively by Alizarin Red staining for calcium (**c**) in a 24-well plate setup compared to medium control. **e, g** Same as **b** but calcium deposition was induced by CPP2- (**e**) and CPP1- (**g**) supplemented medium (50 μg Ca/mL), respectively (calcium deposition in mass units in Ctrl. ranged between 189 and 734 μg Ca/mg protein in **e**). **h** TNFα concentration in human macrophage supernatants after exposing cells to CPP2 (50 μg Ca/mL) and compounds as assessed by immunoassay. High concentrations of IP6, but not $(OEG_2)_2$-IP4, precipitated in the treatment medium as shown in the insets and resulted in significantly increased TNFα levels in supernatants (scale bar = 500 μm). **i** Macrophage viability as determined by the MTS assay compared to medium control. **j, k, l, m** Incubation of severely calcified human femoral arteries (**j, l**) or human aortic valves (**k, m**) with $(OEG_2)_2$-IP4 and without (vehicle) resulted in a reduction of $^{18}$F-NaF uptake in $(OEG_2)_2$-IP4 samples compared with baseline vehicle studies as evident from the maximum average $^{18}$F binding determined by $^{18}$F-NaF-PET imaging (**j, k**) and the maximum intensity projections of collected $^{18}$F-NaF-PET data (**l, m**). All data are expressed as mean ± s.d. $n = 7$ (**j**) and $n = 3$ (**k**) biologically independent samples, and from $n = 4$ (**e**) and $n = 3$ (all others) independent cell experiments. Statistical significance was calculated from an ordinary one-way ANOVA with Tukey's multiple comparisons test or from an unpaired Student's $t$-test to determine significance between two data sets with *$p < 0.05$, **$p < 0.01$, ***$p < 0.001$ vs. gray bar, #$p < 0.001$ vs. CPP1, and §$p < 0.001$ vs. 100 μM $(OEG_2)_2$-IP4.

an adenine and high-phosphate-diet-induced rat model of uremia, in which the compound was administered after serum creatinine levels had increased by fourfold relative to baseline (Fig. 4g, h). The $C_{max}$ after i.v. injection of 10 mg/kg $(OEG_2)_2$-IP4 was almost identical to that of healthy controls (43 ± 12 vs. 35 ± 9 μM) but elimination appeared to be modestly slower in the uremic rats (1.6-fold longer $t_{1/2}$ and twofold longer MRT) (Supplementary Table 4), thus resulting in a higher systemic exposure to the compound (Fig. 4i). Similar results were observed following s.c. administration (Supplementary Table 5). Forty percent of the intact $(OEG_2)_2$-IP4 dose was recovered in the urine of healthy rats, suggesting additional clearance modalities (Fig. 4j). The protein-bound fraction of $(OEG_2)_2$-IP4 in healthy human plasma was 58% and 42%, respectively, calculated over total measured concentrations of 2.8 and 27.5 μg/mL (i.e., 4 and 39.5 μM, respectively), suggesting a potential saturation of the binding sites at high concentrations.

One potential side effect of inositol phosphate-based drugs is toxicity due to chelation of ionized serum calcium. In a previous study, a reduction in serum calcium levels by 17% was observed in uremic rats after 4 h i.v. infusion of 50 mg/kg IP6[45]. No significant lowering of serum calcium was evident after administration of 50 mg/kg $(OEG_2)_2$-IP4 in either the acute or subchronic setting (7 days), measured at the $t_{max}$ of 30 min (Fig. 4k). In a separate in vitro test using human serum, a significant difference in decrease of human serum-ionized calcium levels between $(OEG_2)_2$-IP4 and IP6 became evident starting from 600 μM (Fig. 4l). At 100 μM $(OEG_2)_2$-IP4, 96 ± 2% of control serum-ionized calcium was measured.

The potential off-target activity of 10 μM $(OEG_2)_2$-IP4 was assessed against a panel of 87 enzymes, receptors, or ion channels that are used in standard safety pharmacology screens. The compound did not significantly inhibit or stimulate any of the respective biochemical assays (Supplementary Table 11).

**Efficacy of $(OEG_2)_2$-IP4 in a vitamin D-induced model of VC.** Activity of $(OEG_2)_2$-IP4 was assessed in an animal model of acute and severe calcification induced by vitamin $D_3$ overdose, accompanied by hypercalcemia, hyperphosphatemia, and increased plasma creatinine (Supplementary Fig. 12), and reduced body weight (Supplementary Fig. 13), This model resulted in massive VC, which was observed primarily along the entire aorta (Supplementary Fig. 14) in the tunica media (Supplementary Fig. 15). $(OEG_2)_2$-IP4, dosed s.c. at 30 mg/kg twice daily for 11 days, significantly decreased formation of carotid VC as evidenced from calcium quantification by inductively coupled plasma mass

spectrometry (Fig. 5a), but the decrease in abdominal aorta calcification was not significant (Supplementary Fig. 16). IP6-treated animals had to be sacrificed ahead of treatment schedule on day 10 vs. day 12, due to the appearance of necrotic skin lesions (despite maintaining formulation osmolarity and pH within physiological ranges) (Supplementary Fig. 17). However, calcification levels in IP6-treated animals at day 10 were not lower than those of $(OEG_2)_2$-IP4-treated animals at day 12 (Fig. 5a). Interestingly, in this model $(OEG_2)_2$-IP4 treatment led to a beneficial impact on the renal function of vitamin $D_3$-treated animals, as evident from plasma creatinine levels (Fig. 5c), which occurred in conjunction with a reduction of renal calcium content (Fig. 5b). Creatinine levels were significantly decreased in animals receiving $(OEG_2)_2$-IP4 (39.8 ± 6.6 μM) compared to vehicle controls (59.0 ± 7.9 μM), and remained in the healthy range for rats. Plasma calcium and phosphate levels were not impacted by treatment with $(OEG_2)_2$-IP4 (Supplementary Table 6). Given the severity and resultant variability in calcification burden in this vitamin $D_3$ model, the activity of $(OEG_2)_2$-IP4 was assessed in a modified setup, where VC was induced by oral and less frequent administration of vitamin $D_3$ together with a warfarin-supplemented diet. This model resulted in a robust but milder and less variable induction of VC, and therefore allowed for a better investigation of the $(OEG_2)_2$-IP4 dose–response. Starting from day 1 of the vitamin $D_3$ oral administration, animals were treated with s.c. injections of increasing doses of $(OEG_2)_2$-IP4 administered once or twice daily until day 7. Due to the severe local inflammation induced by the repeated s.c. injection of IP6 in the previous experiment, this control was excluded from the present experiment. In two dosing groups, rats lost significantly more weight than the others compared to day 1, but it was not dose-related (Supplementary Fig. 18). Mortality rate in the vehicle group was 38% (15/40) and declined to 0% (0/18) in the 2 × 25 mg/kg/day dose group, at which the effect was statistically significant (Fig. 5d). $(OEG_2)_2$-IP4 dose-dependently prevented VC in the abdominal aorta, as evident from total calcium measurement (Fig. 5e) and von Kossa staining (Fig. 5f, g). The effect was statistically significant starting at 2 × 25 mg/kg/day and 1 × 12.5 mg/kg/day with regard to total calcium content and % von Kossa-positive area, respectively. Moreover, total calcium (Fig. 5h) and % von Kossa-positive area (Fig. 5i) in the thoracic aorta was significantly reduced starting at 2 × 50 and 2 × 25 mg/kg/day, respectively. Intriguingly, and in contrast to the abdominal aorta, the thoracic aorta did not calcify extensively in all animals, making differences between treatment groups less evident. A significant reduction in bulk calcium in the 2 × 25 mg/kg/day group was also evident in more peripheral arteries, i.e., carotid artery (Fig. 5j) and

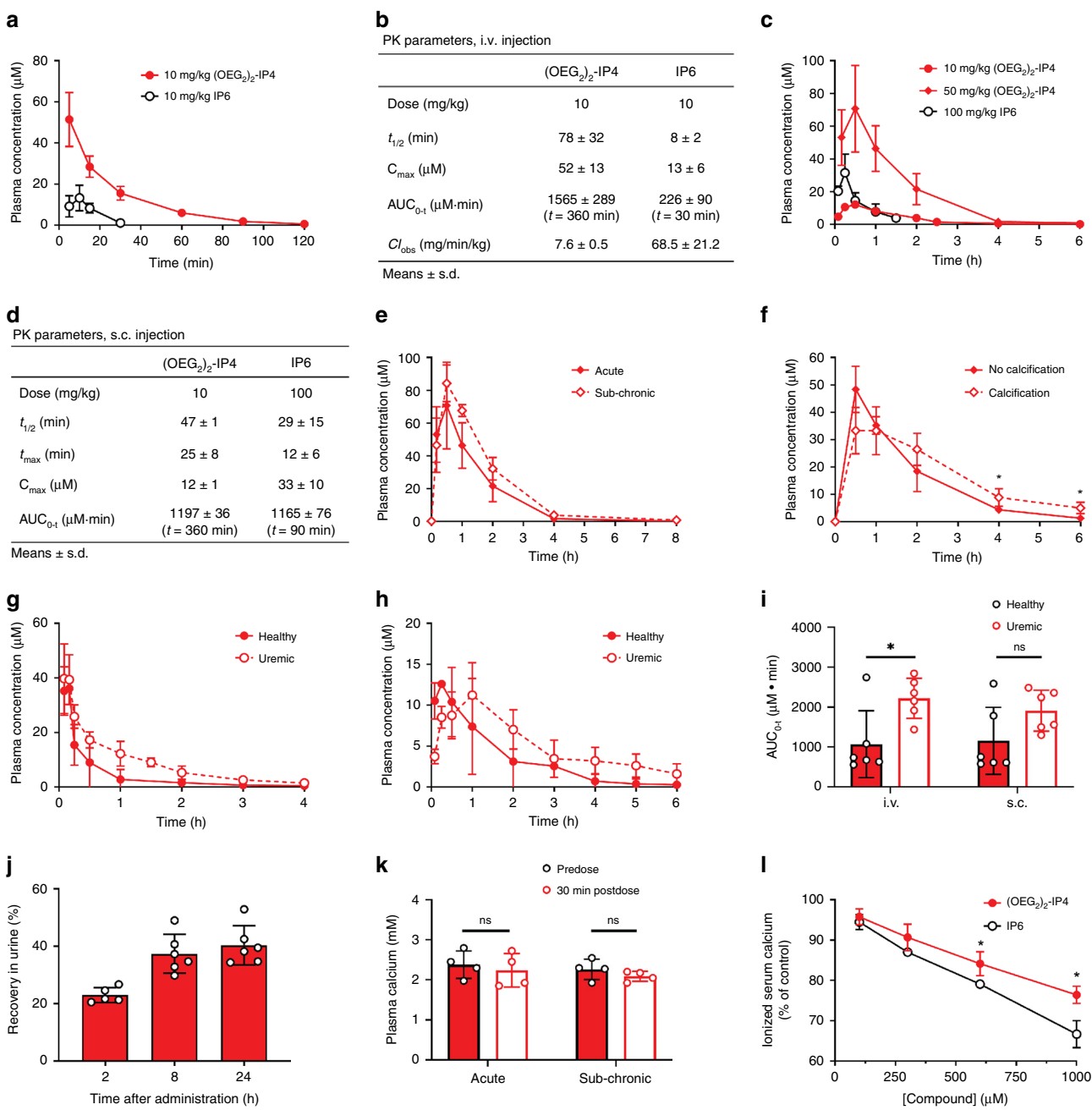

**Fig. 4 PK studies and serum calcium studies with (OEG₂)₂-IP4. a, b** Plasma concentration profiles (**a**) and PK parameters (**b**) of (OEG₂)₂-IP4 (*n* = 3) and IP6 (*n* = 6) in healthy rats following the bolus i.v. injection of 10 mg/kg of compound. **c, d** Plasma concentration profiles (**c**) and pharmacokinetic parameters (**d**) of (OEG₂)₂-IP4 (*n* = 3) and IP6 (*n* = 4) in healthy rats following s.c. administration. Dosing of 10 mg/kg IP6 s.c. resulted in plasma concentrations below the lower limit of quantification (i.e., 772 nM). **e** Plasma concentrations after acute and sub-chronic treatment (7 days of administration) with s.c. injection of 50 mg/kg/day (OEG₂)₂-IP4 (*n* = 4). **f** Plasma exposures following the s.c administration of 50 mg/kg/day (OEG₂)₂-IP4 in a rat model of vitamin D and warfarin-induced VC (*n* = 4). A significant difference was evident after 4 and 6 h. **g, h** Plasma concentration profiles of (OEG₂)₂-IP4 in healthy (non-uremic, *n* = 6) and adenine and high-phosphate-diet-induced calcification rats (*n* = 6) following the i.v. bolus (**g**) and s.c. (**h**) administration of 10 mg/kg of compound. **i** Comparison of the AUC₀₋ₜ of (OEG₂)₂-IP4 after administering into healthy (*n* = 6) and adenine and high-phosphate-diet-induced calcification groups (*n* = 6) with *t* = 360 min for s.c. and *t* = 240 min for i.v. groups. **j** Cumulative recovery of (OEG₂)₂-IP4 from rat urine after s.c. administration of 10 mg/kg (OEG₂)₂-IP4 into healthy rats (*n* = 5 at 2 h, *n* = 6 at 8 and 24 h). **k** Plasma calcium before and 30 min after s.c. administration of 50 mg/kg (OEG₂)₂-IP4 in an acute and sub-chronic (7 days) setting in healthy rats (*n* = 4). **l** In vitro-ionized human serum calcium with (OEG₂)₂-IP4 or IP6 (*n* = 3). All data are expressed as mean ± s.d. from *n* animals. Level of significance was derived from unpaired Student's *t*-test in **l** and non-parametric Mann–Whitney test in **i, k**; *\*p* < 0.05 vs. no calcification (**f**) and IP6 (**l**), respectively; ns = not significant.

femoral artery (Supplementary Fig. 19). Calcification in the heart and the left kidney was low (<0.2 mg Ca/g tissue and <0.5 mg Ca/g tissue, respectively) and no differences between treatment groups were observed. Importantly, (OEG₂)₂-IP4 did not influence the calcemic status of the treated animals. Serum levels of ionized calcium on day 7 were similar after vehicle and (OEG₂)₂-IP4 (50 mg/kg) administration (1.70 ± 0.10 and 1.70 ± 0.13 mM, respectively).

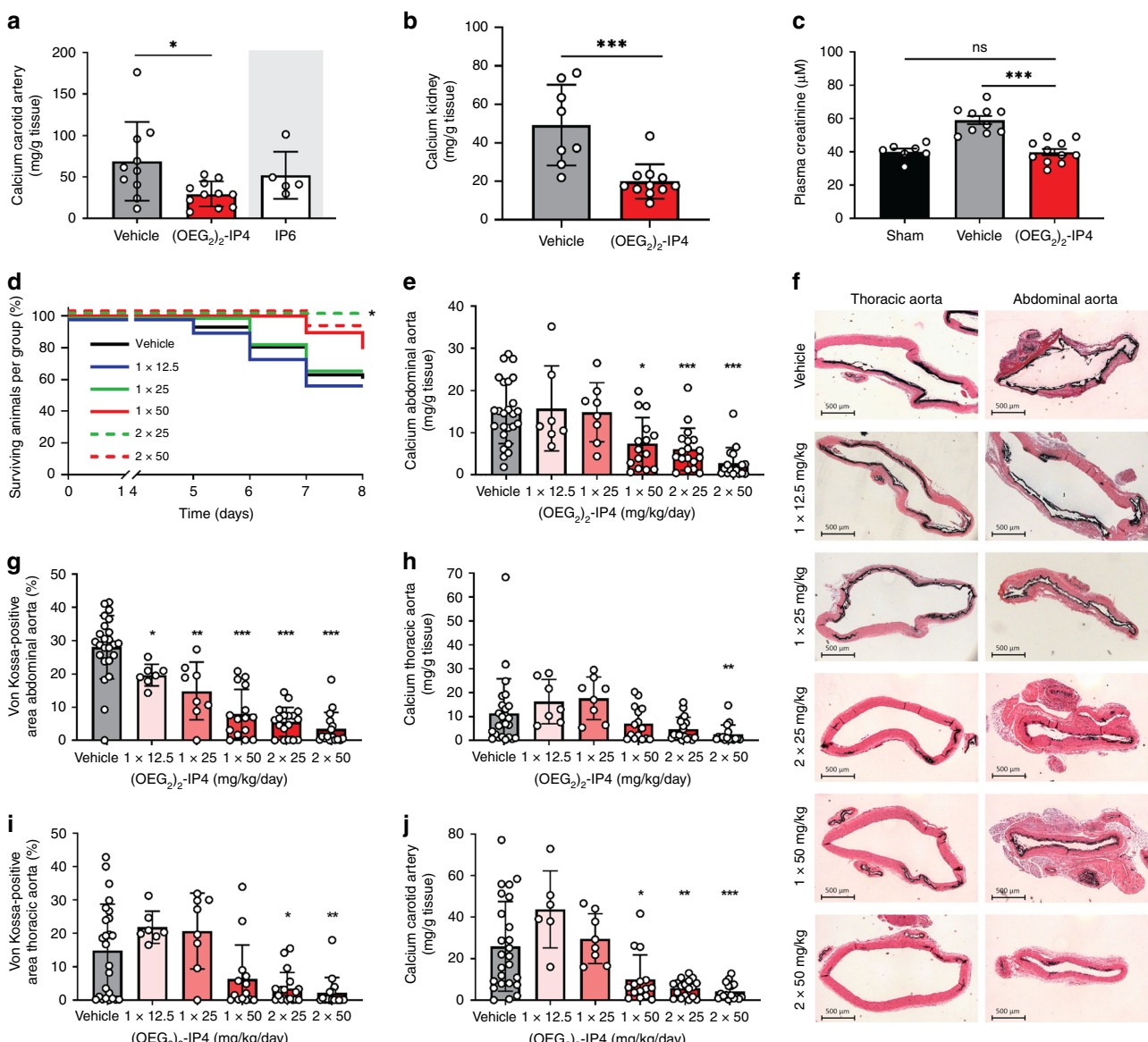

**Fig. 5 (OEG₂)₂-IP4 prevents calcification in a vitamin D-induced rat model of VC.** Calcification was induced by either s.c. administration of 300,000 IU/kg/day vitamin D₃ for 5 days combined with high-phosphate diet (**a–c**) or by oral administration of 100,000 IU/kg/day vitamin D₃ for 4 days combined with a warfarin-supplemented diet (**d–j**). Bulk calcium in the carotids (**a**) and kidneys (**b**), and plasma creatinine (**c**) of vehicle (n = 8 (**b**), n = 10 (**a, c**)), groups treated with twice daily s.c. injections of 30 mg/kg (OEG₂)₂-IP4 (n = 11) or IP6 (n = 5), and sham (n = 7). One rat in the vehicle group died several hours after first vitamin D₃ injection without explanation. Animals in the IP6 group had to be sacrificed 2 days ahead of schedule and no direct comparison can be made. **d** Kaplan–Meier survival plot. **e, g** Calcium quantification (**e**) and von Kossa-positive area quantitation (**g**) in the abdominal aorta. **f** Representative images of aortic calcifications stained with von Kossa (black) (scale bar = 500 μm). Calcium (**h**) and von Kossa-positive area (**i**) quantification in the thoracic aorta. Calcium quantification in the carotid artery (**j**). In **d**, n = 40 for vehicle, n = 12 for 1 × 12 and 1 × 25 mg/kg/day, n = 18 for 2 × 25 mg/kg/day, and n = 19 for 1 × 50 and 2 × 50 mg/kg/day dose groups. In **e–j**, n = 7 (**e–h**) and n = 6 (**j**) for 1 × 12.5 mg/kg/day, n = 8 for 1 × 25 mg/kg/day, n = 14 (**i**) and n = 15 (**e–g, j**) for 1 × 50 mg/kg/day, n = 18 for 2 × 25 mg/kg/day, n = 16 for 2 × 50 mg/kg/day dose groups, n = 24 (**g**) and n = 25 (**e, h–j**) for vehicle. All data are presented as mean ± s.d. from n animals. Statistical significance was derived from non-parametric Kruskal–Wallis and multiple comparison was performed by Mann–Whitney test with Bonferroni correction, with *p < 0.05, **p < 0.01 and ***p < 0.001 vs. vehicle. Comparison of survival was performed by Mantel–Cox test.

**Efficacy of (OEG₂)₂-IP4 in an adenine-induced model of VC.** Uremia increases the risk for developing VC. Therefore, the effect of (OEG₂)₂-IP4 on the development of VC was investigated in a previously established adenine and high-phosphate-diet-induced rat model of VC that results in uremia and chronic renal failure[66]. Renal failure, and subsequent calcification, was induced by feeding a high-phosphate-diet regime for a total of 7 weeks, which also had a low (2.5%) protein content and was supplemented with 0.75% adenine for a period of 4 weeks, starting 2 weeks into the

high-phosphate-diet. (OEG₂)₂-IP4 was administered via a s.c.-implanted osmotic pump at doses of 5, 15, or 50 mg/kg/day for 4 weeks, and treatment was initiated 1 week after the start of adenine supplementation. As opposed to the vitamin D-induced calcification model, mortality in the adenine and high-phosphate-diet-induced VC model was low and did not differ significantly between treatment groups (Supplementary Table 7). Animals in this model developed severe CKD (Supplementary Fig. 20), which was accompanied by substantial weight loss (Supplementary

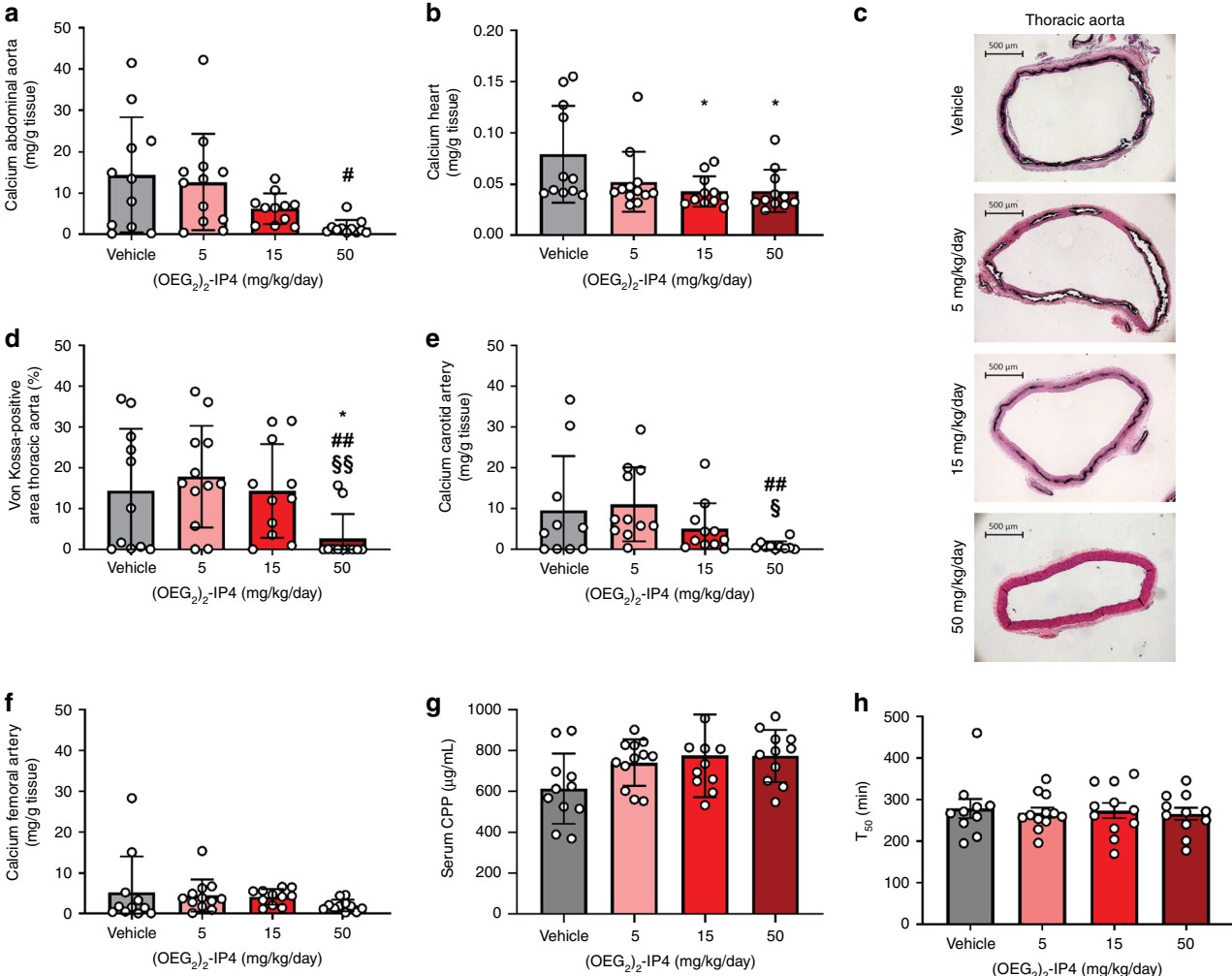

**Fig. 6 (OEG₂)₂-IP4 dose-dependently prevents calcification in an adenine diet-induced rat model of VC. a**, **b**, **e**, **f** Quantification of total calcium in the abdominal aorta (**a**), the heart (**b**), the carotid arteries (**e**), and the femoral arteries (**f**). **c** Representative pictures of calcification in the thoracic aorta stained positive with von Kossa (black) in the different treatment groups (scale bar = 500 μm). **d** Von Kossa-positive area quantitation in the thoracic aorta. **g** Calcium content in CPPs collected from animal sera at sacrifice. **h** $T_{50}$ derived from animal sera collected at sacrifice. Results are presented as mean ± s.d. from $n$ animals, with $n = 11$ for vehicle, $n = 12$ for the 5 mg/kg/day dose group, and $n = 11$ for the 15 and 50 mg/kg/day groups. $N = 10$ for vehicle in **e** and **h**, and for 50 mg/kg/day in **e**. Statistical significance was derived from non-parametric Kruskal–Wallis test and multiple comparison was performed by Mann–Whitney test with Bonferroni correction; *$p < 0.05$ vs. vehicle, #$p < 0.05$, ##$p < 0.01$ vs. 5 mg/kg/day, and §$p < 0.05$, §§$p < 0.01$ vs. 15 mg/kg/day.

Fig. 21). No significant difference in the level of renal failure and body weight loss was observed between the different treatment groups, with the exception of creatinine at week 3 and serum phosphate at sacrifice in the 15 mg/kg/day group (Supplementary Fig. 20).

There was a trend in a dose-dependent reduction of bulk calcium in (OEG₂)₂-IP4-treated animals compared to vehicle in the abdominal aorta, which is especially prone to calcify in this model, with a significant effect when comparing the highest dose (50 mg/kg/day) to the lowest dose (5 mg/kg/day) (Fig. 6a). The inhibitory effect of (OEG₂)₂-IP4 on heart calcifications was significant starting at 15 mg/kg/day (Fig. 6b). The compound's activity was also evident following von Kossa staining (Fig. 6c) on the thoracic part of the aorta, and quantification of the von Kossa-positive area (Fig. 6d). Treatment with (OEG₂)₂-IP4 at 50 mg/kg/day significantly reduced calcification compared to vehicle (2.72 ± 5.97 vs. 14.50 ± 15.11% von Kossa-positive area), and also compared to 5 and 15 mg/kg/day groups (Supplementary Table 8). In line with this, treatment with (OEG₂)₂-IP4 at

50 mg/kg/day also prevented calcification in the carotid artery compared to the 5 and 15 mg/kg/day groups (Fig. 6e). In the femoral artery, reduction in bulk calcium was insignificant for all tested groups, likely due to the low level of calcification detected in the vehicle control (Fig. 6f, Supplementary Table 9). In agreement with the results of the PK study and the vitamin D-induced VC model, the calcemic status of the animals was not influenced by the administration of (OEG₂)₂-IP4 50 mg/kg/day for 4 weeks (Supplementary Fig. 20c). The calcium content derived from isolated serum CPP at sacrifice (Fig. 6g), CPP2 hydrodynamic diameter (Supplementary Fig. 22), and the serum calcification propensity ($T_{50}$) measured in sera collected at sacrifice (Fig. 6h) did not differ between treatment groups.

Tibial bone histomorphometric analysis revealed an unchanged mineralized trabecular bone area for the various (OEG₂)₂-IP4 doses tested (Supplementary Fig. 23). However, a distinct increase in the osteoid area was observed in the treatment groups, which, in the presence of an intact osteoblast number/activity, points towards an imbalance between osteoid deposition and subsequent

mineralization (Supplementary Fig. 24). As both osteoblast and osteoclast morphology were not impacted (Supplementary Fig. 25), this observation suggests that $(OEG_2)_2$-IP4 may not exert cellular toxicity, and that the effect on mineralization of growing bone is probably due to a passive adsorption effect. In order to investigate the 3D bone structure, micro-computed tomography (μCT) analysis was performed, revealing the mineralized trabecular bone structures to be better preserved in the $(OEG_2)_2$-IP4-treated animals compared to vehicle-treated animals (Supplementary Fig. 26, Supplementary Table 10). The low volume of distribution of $(OEG_2)_2$-IP4 observed in the PK study also suggests that only little compound leaves the circulation, and hence is available to interact with bone.

**Discussion**. Cardiovascular disease remains the leading cause of death in the world despite significant advances in the development of therapies addressing classical risk factors such as high cholesterol, hypertension, and diabetes, combined with improved awareness of the negative impacts of obesity and smoking. This highlights the importance of investigating different strategies to combat cardiovascular diseases. One largely unexplored target for pharmacological intervention is VC, despite the fact that it has consistently been linked to poor clinical outcomes[8]. In fact, the coronary calcification score is the strongest predictor of coronary events in asymptomatic patients[67]. Certain high-risk populations particularly prone to VC have been identified, such as CKD patients for which the extent of VC and risk of cardiovascular disease increase dramatically as renal function declines[68].

In the present study, a series of inositol phosphate derivatives were examined for their potential to inhibit VC. Screening for modifications both in the number and size of OEG moieties on *myo*-inositol phosphates led to the identification of $(OEG_2)_2$-IP4 as a potent inhibitor of VC. $(OEG_2)_2$-IP4 displayed activity in the high nanomolar range in the competitive $^{18}F$-NaF CPP-binding assay, and prevented the growth of nascent CPPs, their transition into the crystalline phase, and calcification of VSMC monolayers in the low micromolar range. Conversely, IP6 was associated with mid-micromolar activity or a lack of dose–response in these assays. As opposed to $(OEG_2)_2$-IP4, IP6 did not stabilize CPP in their primary state and led to the formation of CPP2s, which are hypothesized to initiate pathological response[29,30].

In the case of the CaP-induced VSMC calcification model, the $(OEG_2)_2$-IP4 possibly acts in multiple ways to prevent calcification. Matrix vesicles secreted by VSMCs under pro-calcifying conditions form the first nidus for mineralization and chemisorption of $(OEG_2)_2$-IP4 onto such nidi may result in crystal growth inhibition[57]. Potential intracellular effects, although unlikely due to the compound's hydrophilicity and charge making it difficult to diffuse through cell membranes, cannot be excluded and should be further investigated.

$(OEG_2)_2$-IP4 was efficacious in both adenine and high-phosphate-diet-induced, and vitamin D-induced rat models of VC, and displayed dose–response in vitro as well as strong in vivo activity without affecting serum calcium levels. Importantly, $(OEG_2)_2$-IP4 exhibited superior activity and prolonged plasma exposure compared to IP6 following bolus i.v. and s.c. administration, the latter of which opens the possibility of therapeutic use in a chronic and ambulatory setting. The PK data show that after s.c. injection plasmatic concentrations above the $IC_{50}$ were achieved for at least 4 h vs. 12 min for IP6 (Fig. 4c), which partly explain the higher efficacy achieved with $(OEG_2)_2$-IP4. Moreover, to our knowledge, this is the first study showing that a pharmacological treatment can impact the $^{18}F$-NaF-PET signal in the context of VC, and it may therefore serve as a useful tool for pharmacodynamic read-outs in pre-clinical and early stage

clinical trials, given the growing evidence of its value as a predictive marker for calcification progression[33,34,69].

The differences in activity between $(OEG_2)_2$-IP4 and IP6 are unlikely to be mediated solely by steric hindrance of the OEG moieties, as the ethylene glycol chain length is short compared to what is typically used for nanoparticle coating (0.1 vs. 2–5 kDa)[47]. The observation that IP6 is a stronger chelator of soluble ionic calcium than $(OEG_2)_2$-IP4, but a weaker inhibitor of calcification, leads to the hypothesis that an equilibrium exists between binding to soluble and solid forms of calcium. Therefore, the reduced charge density of $(OEG_2)_2$-IP4 may decrease its binding to soluble ionic calcium (present in large excess in biological fluids) rendering it more available for binding to solid forms of calcium. The OEG chains may prevent stacking and precipitation of inositol phosphate–calcium complexes and hence further contribute to reducing mineral accretion.

This mechanism of action is possibly similar to that of extracellular matrix and serum proteins that contain acidic and phosphorylated residues in order to function as exquisite regulators of biomineralization[32,70]. Our observation is in line with the structural basis of calcification inhibition by fetuin-A, where surface binding of calcium is mediated by four negatively charged amino acids on the extended β-sheet of domain D1 that occupy phosphate positions on the face of apatite crystals[71].

It can be hypothesized that $(OEG_2)_2$-IP4 acts by stabilizing CPP2 particles and reducing their uptake by VSMCs and subsequent calcium overload, which may cause downstream procalcific and pro-inflammatory events[24,30]. Chronic exposure to CPPs, e.g., when clearance becomes overwhelmed as in CKD, is a likely driver of VC[24]. CPPs were found to be present in serum from high-phosphate-diet-fed rats with adenine-induced renal failure[72]. By stabilizing CPPs, $(OEG_2)_2$-IP4 may aid mineral clearance from the circulation before VC can ensue. No delay in $T_{50}$ was observed when serum samples were tested from the adenine-induced VC rat model, as $(OEG_2)_2$-IP4 was likely to be already eliminated or bound to existing calcified deposits at the time of sampling.

Although PEGylation of small molecules with relatively long PEG chains (>2 kDa) has previously been reported to improve PK, it was unexpected to discover that conjugation with only two ethylene glycol repeating units was sufficient to increase systemic exposure[73]. The prolonged plasma half-life of $(OEG_2)_2$-IP4 compared to IP6, observed in healthy animals, is likely due to the enhanced metabolic stability of $(OEG_2)_2$-IP4. Forty percent of the intact compound was recovered in the urine, suggesting additional routes of elimination, such as hepatic clearance and degradation in plasma that may result in dephosphorylated metabolites of $(OEG_2)_2$-IP4. Plasma exposure of $(OEG_2)_2$-IP4 was somewhat extended in adenine and high-phosphate-diet-fed animals compared to healthy controls. This effect is seen for many active pharmaceutical ingredients, and can be rationally explained by the decreased renal clearance. In contrast, the plasma level of IP6 has been reported to be decreased in uremic rats[45], possibly due to increased enzymatic metabolism in the state of uremia, which for $(OEG_2)_2$-IP4 may be counterbalanced by its higher stability.

Since the OEG polymer shows little toxicity, is highly water soluble and eliminated from the body intact via the kidneys (PEGs < 30 kDa), and IP6 has previously been administered to patients displaying acceptable safety and tolerability[43], the likelihood of observing serious toxic side effects with this chemical entity is low.

In the present study, we observed an effect of $(OEG_2)_2$-IP4 on bone osteoid area. However, it was seen in a CKD/VC rat model in young animals with open growth plates and rapid bone growth that presents a dramatically increased bone turnover with chaotic

bone mineralization, making the rats particularly vulnerable to an imbalance between excessive osteoid deposition, subsequent mineralization, and bone resorption[74]. Therefore, this model does not accurately reflect the low to mildly increased bone metabolism in the majority of CKD patients[75]. On the other hand, (OEG$_2$)$_2$-IP4 treatment prevented trabecular bone loss inherent to CKD as could be deduced from 3D-structure analysis of mineralized bone. This may be due to inhibition of osteoclastic adhesion to mineralized bone, which in turn would result in reduced osteoclastic activity. Further experiments are required to investigate the effect of (OEG$_2$)$_2$-IP4 on bone mineralization.

In summary, this study identified (OEG$_2$)$_2$-IP4, a functionalized *myo*-inositol phosphate and provided early proof-of-concept on the applicability of (OEG$_2$)$_2$-IP4 as a potentially universal inhibitor of pathological soft tissue calcification. Our data strongly support a mechanism of action in which growth of VC is hampered by electrostatic interactions of (OEG$_2$)$_2$-IP4 at the level of nano-, micro-, and macrocalcifications, while preserving calcium and phosphate homeostasis. Based on its mechanism of action, (OEG$_2$)$_2$-IP4 may hold great promise in directly targeting the end product of the VC process (i.e., hydroxyapatite growth and deposition) in a range of disorders in which VC leads to impaired cardiovascular function. Ultimately, this work warrants the further development of (OEG$_2$)$_2$-IP4 (i.e., INS-3001)[76,77] in order to generate clinical evidence of its therapeutic potential for the benefit of patients.

## Methods

**Compounds**. Synthesis of (±)-OEG$_{12}$-IP5, (OEG$_2$)$_2$-IP4, and (OEG$_7$)$_3$-IP3 are described in detail in the Supplementary Information 3.1. (OEG$_2$)$_3$-IP3 as bis-magnesium salt and (OEG$_2$)$_2$-IP4 as tetra-sodium salt were purchased from Chimete Srl (Tortona, Italy). All other oligoethyleneglycol (OEG) derivatized inositol phosphates as listed in Supplementary Table 12 were purchased as sodium salt from Biosynth AG (Thal, Switzerland). Identify and purity was verified by $^1$H, $^{31}$P-NMR (Appendix NMR Spectra), $^{13}$C NMR, and $^1$H,$^1$H-DQF-COSY and ESI-MS (Appendix ESI-MS Spectra).

Control compounds were *myo*-inositol hexakisphosphate (IP6, dodecasodium salt, P-6700, Biosynth AG), Ins(1,2,3,5,6)P$_5$ (IP5, 10008453), Ins(2,3,5,6)P$_4$ (IP4, 10008455), Ins(1,4,5)P$_3$ (IP3, 10008205), and Ins(2,4)P$_2$ (IP2, 10008419) (sodium salts, Cayman Chemical, Ann Arbor, MI, USA), *myo*-inositol (57570, Fluka Analytical, Buchs, Switzerland), alendronate (A2456), and etidronate (D4159) (TCI Deutschland GmbH, Frankfurt, Germany), *myo*-inositol trispyrophosphate (ITPP, I666020, Toronto Research Chemicals, Toronto, ON, Canada), sodium pyrophosphate (PP$_i$, AB118827, ABCR, Karlsruhe, Germany), magnesium citrate (63067, Fluka Analytical), and PEG-alendronate, which was synthesized as described previously[48].

**In vitro screening assay**. The screening assay was performed as described previously[22]. In brief, pooled human serum (H4522, Sigma-Aldrich) stored at −20 °C was thawed at 37 °C, centrifuged (10,000 × *g*, and 4 °C, 30 min) (5427R, Eppendorf, Switzerland), and sterile filtered (0.22 μm filter, Millipore, Merck KGaA) to remove potential cryoprecipitates (cleared human serum). The calcium solution was 40 mM CaCl$_2$ (102382, Merck KGaA) in HBS, pH 7.4 (RT) (100 mM HEPES (H3375, Sigma-Aldrich) and 140 mM NaCl (1734712, Fisher Scientific, Hampton, NH, USA)). The phosphate solution contained 24.00 mM PO$_4$ resulting from the combination of 19.44 mM Na$_2$HPO$_4$ (28026.292, VWR Chemicals, Radnor, PA, USA) and 4.56 mM NaH$_2$PO$_4$ ·2H$_2$O (106346, Merck KGaA) in HBS, pH 7.4 (RT). All salt solutions and buffers were double sterile filtered (0.22 μm) and pre-warmed to 37 °C prior to use. Calcium solution, serum, phosphate solution, and test compounds in Milli-Q water were mixed together at a volume ratio of 5:8:5:2 in a 96-well plate (Costar 96-well plate, clear, flat bottom, 3370, Corning Life Sciences, Corning, NY, USA) with 1 min shaking after each addition to a total volume of 200 μL per well. This resulted in a final concentration of 10 mM calcium and 6 mM phosphate. The plate was sealed with an adhesive film and the absorption at 550 nm was measured every 3 min with 1 min in-between shaking for a total of 300 cycles at 37 °C using the infinite M200 microplate reader (Tecan, Männedorf, Switzerland). An $A_{550}$ vs. time graph was generated and $T_{50}$ (min) (time at half-maximum transition from CPP1 to CPP2) values were derived by sigmoidal curve fit using Prism 8 (nonlinear regression in the log(agonist) vs. response–variable slope (four parameter mode) using the robust fit fitting method, GraphPad Software, San Diego, CA, USA). Activities between compounds were compared by generating a $T_{50}$ (min) vs. concentration (μM) graph, calculating nonlinear regression and deriving the concentration required to delay $T_{50}$ to 350 min (c350) using a standard equation for the dose–response curve, Eq. (1), whereby $x =$

[compound], $y =$ time (min), bottom = baseline response, top = maximum response, $T_{50} =$ [drug] that leads to half-maximum response, and the hill slope is a factor that describes the steepness of the curve.

$$y = \text{bottom} + \frac{(\text{top} - \text{bottom})}{1 + \left(\frac{T_{50}}{x}\right)^{\text{hillslope}}}. \quad (1)$$

Next, (OEG$_2$)$_2$-IP4, (OEG$_{11}$)$_2$-IP4 and IP6 at c350 concentration (5.1, 3.8, and 32.9 μM, respectively) were incubated in enzymatically active human serum, used within 24 h after collection, at 37 °C for 0 and 4.5 h. Human serum was purchased from Blutspende Zurich (Schlieren, Switzerland), following a formal application for order of products for laboratory and research purposes, based on the valid Swiss law of Human Research (HMG). Subsequently, samples were assayed in the screening assay (vide supra). $T_{50}$ values were derived and the decrease of $T_{50}$ ($\Delta T_{50}$) of a sample incubated in enzymatically active serum for 4.5 h in comparison to 0 h was reported. Four independent assays using purchased serum from four healthy human donors were performed.

**Phytase assay**. The enzyme 3-phytase from *Aspergillus niger* was purified from Natuphos® (BASF, Ludwigshafen, Germany) as described previously[78] and aliquots at 4.64 mg/mL were stored at −80 °C until use. Test compounds at 15 μM were incubated with 23.2 μg/mL of 3-phytase in 10 mM Tris (pH 7.4, RT) (T1503, Sigma-Aldrich) at 37 °C. Phosphate hydrolysis was monitored over 24 h by spectrophotometric quantification of hydrolyzed phosphate using the ammonium molybdate malachite green method as described previously[79].

**Calciprotein particles**. In vitro CPPs were prepared as described previously[61]. In brief, saline (140 mM NaCl), cleared human serum, phosphate solution, and calcium solution were combined at a volume ratio of 2:8:5:5 to a final volume of 30 mL in a 50 mL falcon tube (TPP, Trasadingen, Switzerland) with mixing by inversion after each addition. All salt solutions and buffers were double sterile filtered (0.22 μm) and pre-warmed to 37 °C prior to use. The mixture was incubated with continual slow mixing on a rotator (240 rotations/min, KS 130 basic orbital shaker, IKA, Germany) at 37 °C (WTB Binder, Germany) for 1 h for CPP1 or 14 h for CPP2 formation, respectively. Subsequently, 2-mL aliquots were subjected to centrifugation (20,000 × *g*, 4 °C, 2.5 h). The resultant pellets were washed twice with ice-cold HBS, and resuspended by thorough pipetting in 200 μL HBS at 37 °C. The CPP suspensions were pooled and centrifuged (1000 × *g*, RT, 10 min) to pellet large aggregates, and the supernatant (CPP suspension) was collected. An aliquot thereof was dissolved in 0.1 M HCl and CPP calcium was quantified using a calcium colorimetric assay (MAK022, Sigma-Aldrich) and adjusted to 1 mg Ca per mL with 100 mM HBS (pH 7.4 (RT)). The CPP suspensions were either used directly or were snap frozen in liquid nitrogen and stored at −80 °C until further use and up to a maximum of 2 months, during which CPP stability was confirmed by DLS.

**TEM and selected area electron diffraction**. CPPs were prepared (vide supra) and TEM was performed similar to previous reports[71]. Additionally, CPP samples were prepared plus (OEG$_2$)$_2$-IP4 or IP6. Compounds were added at a final concentration of 100 μM after combining mineral solutions and serum to initiate CPP formation and the mixture was incubated for 14 h. Herein, cell culture grade 50 mM HEPES (1 M HEPES, 5630056, Thermo Fischer Scientific) was used for washing steps, to avoid high solvent and particle background in TEM. In brief, CPPs at 0.5 mg calcium/mL were drop-casted onto a copper grid, 300 mesh, covered with a continuous 8-nm-thick carbon film. The TEM data were collected with an FEI Tecnai F30 FEG TEM operated at 300 kV acceleration voltage and equipped with a Gatan 749 MultiScan CCD camera (Thermo Fischer Scientific). The bright-field images were collected at ×39,000 magnification using a 10-μm objective aperture. To assess the crystallinity of the sample selected area diffraction patterns were collected in parallel at a camera length of 300 mm. The radii of the rings in the diffraction pattern were measured (1/nm) and the associated interplanar *d*-spacings (nm) were calculated.

**X-ray powder diffraction**. CPPs were prepared as described above. Herein, cell culture grade 50 mM HEPES was used for washing steps to avoid NaCl background signals. Neat CPP pellets were analyzed on a Stoe IPDS II (Stoe & Cie, Darmstadt, Germany) with an image plate detector, Cu-*K*α radiation ($\lambda = 1.54186$ Å), graphite monochromator, fiber optics, and a modified sample holder (in-house) for transmission powder diffractometry.

**Dynamic light scattering**. The particle hydrodynamic diameter and polydispersity index of CPPs were assessed by DLS of samples diluted to 0.5 mg Ca/mL in 100 mM HBS (pH 7.4, RT) in a total volume of 500 μL at 25 °C using a DELSA$^{\text{TM}}$ Nano C Particle Analyser (Beckman coulter, Brea, CA, USA). All size measurements were carried out at a fixed scattering angle of 165° and the cumulant method was used to report the measured particle size.

Human serum studies were performed under sterile conditions at 37 °C as described previously[29]. In brief, pre-warmed (37 °C) solutions of saline, cleared serum, phosphate solution, and calcium solution were combined under sterile

conditions in a volume ratio of 2:8:5:5 to induce CPP formation. For the pre- and post-setup, $(OEG_2)_2$-IP4 or IP6 in Milli-Q water were added to a final concentration of 3, 10, 30, and 100 μM before and after the addition of phosphate and calcium solution, respectively. For serum background measurements, saline, human serum, and 100 mM HBS (pH 7.4 (RT)) were combined in a volume ratio of 2:8:10. For compound background measurements, $(OEG_2)_2$-IP4 or IP6 in Milli-Q water were added to a final concentration of 30 and 100 μM to the serum background samples. The mixtures were incubated with continual slow mixing on a rotator (240 rotations/min, IKA KS 130 Basics Orbital Shaker) at 37 °C. After 1, 3, and 5 h of incubation, 500 μL aliquots were drawn under sterile conditions to prevent microorganism growth contamination and subjected to DLS analysis at 37 °C. For each experiment, three replicates were produced and a single measurement per sample and time point was performed.

DLS analysis of CPPs induced in serum samples collected from in vivo adenine-induced VC studies and stored at −20 °C was performed as described for human serum samples and CPP hydrodynamic diameter development was monitored by hourly DLS measurements.

In all DLS experiments, the cumulant hydrodynamic diameter and cumulant polydispersity index from three consecutive measurements was presented unless stated otherwise.

**Cell cultures**. All cells were cultured at 37 °C in a humidified atmosphere with 5% $CO_2$ and were tested for mycoplasma contamination after putting cells into culture and subsequently monthly using the MycoAlert™ Mycoplasma Detection Kit (Lonza, Basel, Switzerland). All cell culture experiments in 24- and 96-well plates (TPP Tissue Culture Test Plate, Techno Plastic Products AG, Trasadingen, Switzerland) were performed in two- and six-well replicates, respectively, and were repeated independently three times using different batches of CPP2 and test compounds, unless stated otherwise.

Primary human VSMCs (C-12533, PromoCell, Heidelberg, Germany) were cultured according to the manufacturer's guidelines in SMC-2 culture medium (C-22062, PromoCell) and used up to passage 7. For calcification experiments cells were seeded onto 24-well plates (TPP) at a density of 20,000 cells/well and left to attach for 48–72 h. Complete medium was M199 medium (31150022, Thermo Fischer Scientific), supplemented with 10% v/v FBS (10270106, Thermo Fisher Scientific, Waltham, MA), 1% v/v penicillin/streptomycin (15140122, Thermo Fisher Scientific), and 4 mM L-glutamine (25030024, Thermo Fisher Scientific)[80]. Calcification of cell monolayers was induced as described previously[81] by treatment with CPP2-supplemented (50 μg Ca/mL) medium with or without test compounds (1, 3, 10, 30, and 100 μM) at a volume of 1 mL/well for 48 h at 37 °C, 5% $CO_2$. In a second assay, calcification was induced similar to previous experiments[56] by cell treatment with or without test compounds (0.01, 0.1, 1, 10, and 100 μM) in complete medium supplemented with calcium (2.7 mM) and phosphate (2.5 mM) (CaP medium) at a volume of 1 mL/well for 5 days, and treatment medium was replaced on alternate days.

Potential particle growth and formation of precipitate in CaP medium alone (without cells) was assessed by incubating 1 mL of complete medium, CaP medium, and CaP medium supplemented with 30 μM $(OEG_2)_2$-IP4, respectively, in 24-well plates at 37 °C, 5% $CO_2$ for 48 h. At 0, 12, 24, 36, and 48 h, samples were collected and subjected to pH measurement (pH Elektrode InLab Micro, Mettler Toledo, Giessen, Germany), DLS analysis (500 μL of undiluted sample, measured at 25 °C), and absorbance measurement at 570 nm. Furthermore, microscope images were taken at 0 and 24 h (CTR6000, Leica Camera AG, Wetzlar, Germany) using the following settings: TL-BF, ×25 magnification, A1.00 magnification changer, exposure time 1.5 ms, 37 °C.

Calcium deposition on cell monolayers was assessed qualitatively by staining for calcium with 2% w/v Alizarin Red S aqueous solution (pH 4.2) (05600, Fluka Analytical). Cell monolayers in 24-well plates were washed with PBS (10010015, Thermo Fischer), fixed in 10% v/v NBF in PBS (F8775, Sigma-Aldrich) for 45 min at RT, washed with milli-Q water, incubated for 3 min with 300 μL Alizarin Red S solution, washed with milli-Q, and images were acquired using a Axiovert 35 microscope (Zeiss, Carl Zeiss Microscopy GmbH, Jena, Germany) at ×100 magnification. Calcium quantitation was performed after washing cells with PBS and then decalcifying cell monolayers with 0.1 M HCl overnight at 4 °C. Acidic supernatants were centrifuged (10,000 × g, 4 °C, 10 min) and calcium content was assessed using the calcium colorimetric assay. After decalcification, cell monolayers were solubilized (0.1 M NaOH and 0.1% w/v sodium dodecyl sulfate (71729, Sigma-Aldrich)), and protein content was measured using the Micro BCA Protein Assay Kit (23235, Thermo Fischer).

Occurrence of precipitation in the different treatment groups was assessed visually by the naked eye and by a microscope (CTR6000, Leica Camera AG) at ×25 magnification at 0 and 24 h (performed once in duplicate).

Human monocytes THP-1 (TIB-202™, ATCC, Manassas, VA, USA) were cultured in suspension in Roswell Park Memorial Institute 1640 medium (RPMI GlutaMAX™, 72400021, Thermo Fisher Scientific) supplemented with 10% v/v FBS (complete medium). Cells were passaged according to the manufacturer's guidelines and used up to passage 7. For macrophage differentiation, medium was further supplemented with 100 nM phorbol 12-myristate 13-acetate (PMA, P8139, Sigma-Aldrich) and THP-1 cells were then treated for 3 days at 37 °C, 5% $CO_2$. Subsequently, cells were left to rest in complete medium for 24 h before

experimental use. This method was based on previous work[82] and concentrations of 10–200 nM PMA were screened in order to identify the minimum concentration necessary to obtain a robust differentiation. Differentiation was assessed by observation of morphology under a bright-field microscope (CTR6000, Leica Camera AG) at ×200 magnification, and by cell viability as a surrogate for macrophage attachment to the well bottom.

THP-1 cells were plated at a density of 200,000 cells/well in a 24-well plate, differentiated as stated above, and subsequently treated with or without CPP2 (50 μg Ca/mL) in complete medium with or without test compounds (3, 10, 30, 100 μM) for 24 h at 37 °C, 5% $CO_2$.

Occurrence of precipitation in the different treatment groups was assessed visually by the naked eye and by a microscope (CTR6000, Leica Camera AG) at ×25 magnification at 0 and 24 h (performed once in duplicate).

The cell culture supernatants from the cells treated as described above were collected at the end of the respective treatment. Supernatants were harvested in 1× cOmplete™ protease inhibitor cocktail (Roche, Basel, Switzerland), centrifuged (1000 × g, 4 °C, 10 min), snap frozen in liquid nitrogen, and stored at −80 °C until use.

Human TNF-α was assayed in cell culture supernatants using commercial multiplex immunoassay kits (U-Plex of TNF-α (custom-made multiplex plate) and Human TNF- Kit (K151QWD-1), both from MesoScaleDiscovery, Rockville, MD, USA), and corrected for total protein content using the Micro BCA™ Protein Assay Kit.

Cell viability was assessed using the MTS assay (CellTiter 96® Aqueous One Solution Cell Proliferation Assay, G3580; Promega, Madison, WI, USA) according to the manufacturer's instructions. Assays using combined treatment of CPPs or CaP and test compounds were performed in 24-well plates using 20,000 cells per well and assays using test compounds only were performed in a 96-well plate format with a seeding density of 10,000 (VSMC) or 50,000 (THP-1) cells per well, respectively. For controls, cells were treated with respective complete medium only and results were calculated as % of control.

Triplicate aliquots of CPP2s (20 μg Ca/mL) in 100 μL assay buffer (100 mM HBS, pH 7.4 (RT)) were incubated for 1 h at RT in the presence of increasing concentrations of compounds in 50 μL Milli-Q water and $^{18}F$-NaF (1 nM) in 50 μL assay buffer, resulting in a total incubation volume of 200 μL ($^{18}F$-NaF was prepared using a FASTlab synthesizer (GE Healthcare, Little Chalfont, UK) and commercially available $^{18}F$-NaF cassettes (GE Healthcare)). Maximum radiotracer binding was determined by incubating CPP2s with $^{18}F$-NaF (1 nM) only. The reaction was terminated by rapid filtration through a SOLAμ™ SPE plate (HRP 2 mg/mL 96-well plate, 22 μm; Thermo Fisher Scientific). SPE individual column matrices were washed twice rapidly in assay buffer and radioactivity on the filters determined by an automatic gamma counter (Wizard2 Gamma Counter, Perkin Elmer, Waltham, MA, USA). Percentage binding was calculated as maximum binding of $^{18}F$ (absence of $(OEG_2)_2$-IP4 or IP6) minus measured binding of $^{18}F$ in presence of different compound concentrations divided by maximum binding and multiplied by 100. The compound concentration, where the percentage binding is reduced by half ($IC_{50}$) was determined by single site-specific binding model with a Hill slope of 1. $IC_{50}$ is the concentration of agonist that gives a response half way between the bottom and top parts of the curve. $^{18}F$-NaF micro-PET/computed tomography (micro-PET/CT) was employed for ex vivo analysis of $(OEG_2)_2$-IP4's engagement to target calcified human tissue explants, as described previously[54]. In brief, calcified human tissue samples were incubated with $(OEG_2)_2$-IP4 ($IC_{90}$ = 8554 nM) or vehicle (water) in 200 mL of human serum (H4522; Sigma-Aldrich) for 30 min at RT. Subsequently, $^{18}F$-NaF was added (100 kBq/mL, equivalent to approximately 1 nM, derived from previous CPP2 studies where $K_D$ of $^{18}F$-NaF to CPP2s was 1 mM) to the incubation medium and the mixture was incubated for 30 min at RT. Following incubations, samples were washed twice in 0.1 M PBS (1 min per wash) and mounted on a Petri dish for micro-PET/CT imaging (nanoPET/CT, Mediso, Budapest, Hungary). In total, seven samples from the femoral arteries of three patients with ischemic peripheral disease and three samples from native aortic valves of three patients undergoing valve replacement surgery were used. All samples were collected at the time of surgery and stored in 4% paraformaldehyde at RT until further use. All patients gave their written, informed consent. All procedures had local ethical approval (06/S0703/110, 12/WS/0227, 09/S0703/118, and 12/NW/0036). All studies were approved by East and West Scotland Research Ethics Committees, and all experiments were conducted according to the principles expressed in the Declaration of Helsinki.

**Pharmacokinetics**. All PK studies in healthy animals were performed by the contract research organization Aphad S.r.l. (Milan, Italy). In all studies, compounds were administered as a salt, and the doses and plasma quantitation refer to the acid form. PK parameters listed in Supplementary Table 13 were derived from measured plasma concentrations applying noncompartmental analysis. Concentration data were extrapolated using the software Analyst™ 6.1 (Applied Biosystems, Foster City, CA, USA), the area under the plasma concentration vs. time curve (AUC) was calculated by linear trapezoidal rule using PK solver 2.0 add-in for Microsoft Excel 2007 (available for download in[83]), and a uniform weight was performed as a first general approach.

To study PK in healthy animals, male Sprague-Dawley rats (Harlan, Correzzana, Italy) were housed up to three per cage with free access to water and

standard chow (pellets 12 mm, global diet 2018 certificate, Mucedola, Srl, Settimo Milanese, Italy), and submitted to a 12-h light/dark cycle. Animals were administered i.v. 10 mg/kg of $(OEG_2)_2$-IP4 in saline solution i.v. (30 s bolus, 5 mL/kg) and s.c. (5 mL/kg), respectively, in three animals per route and treatment. Animals were administered 10 mg/kg IP6 i.v. (30 s bolus, 5 mL/kg) in six animals, and s.c. (5 mL/kg) at 10 and 100 mg/kg in three animals per treatment. PK after acute and sub-chronic application was studied using four animals per group dosed with s.c. (5 mL/kg) 50 mg/kg $(OEG_2)_2$-IP4 (in saline) a single time or once daily for seven consecutive days, respectively. Plasma calcium concentration was determined in pre-dose samples and in $t = 30$ min samples of both acute and chronic treatments using the calcium colorimetric assay. Blood samples of 100 μL were drawn at several time points from the caudal vein, centrifuged at $(3000 \times g, 4\,°C, 10 \text{ min})$ and stored at $-80\,°C$ until analysis by liquid chromatography–tandem mass spectrometry (LC-MS/MS).

To study urinary excretion, six animals were administered s.c. 10 mg/kg of $(OEG_2)_2$-IP4 and two animals served as blank urine controls. Animals were placed in separated metabolic cages (TECNIPLAST S.p.A. Buguggiate, Italy) right after dosing, left there for the entire experiment, and urine was collected after 2, 5, 8, and 24 h. At the end of the experiment, cages were washed with 10 mL water and the washes were collected in separated tubes. Urine samples were centrifuged at $3000 \times g$ and $4\,°C$ for 10 min and stored at $-20\,°C$ until quantification of $(OEG_2)_2$-IP4 by LC-MS/MS. Recovery (%) was calculated using Eq. (2).

$$Recovery(\%) = \frac{(OEG_2)_2IP4(\mu g)}{dose(\mu g) \times rat\ weight(kg)} \times 100. \qquad (2)$$

**Animals with induced VC**. To study PK in rats presenting with VC, plasma samples were used from a sub-group of the study where calcification was induced with vitamin D and warfarin treatment. Herein, four animals with manifest calcification were given s.c. 50 mg/kg $(OEG_2)_2$-IP4 (5 mL/kg) on day 1 and day 7, with no drug administration on intermediate days, and plasma samples for PK analysis were collected and stored at –80 °C until drug quantitation by LC-MS/MS.

To study PK in uremic animals, male Sprague-Dawley rats (Charles River Laboratories, L'Arbresle, France) were housed two per cage with free access to water and standard chow (rat/mouse maintenance diet, SSNIFF Spezialdiäten GmbH, Soest, Germany) and submitted to a 12-h light/dark cycle. Animals were fed a low-protein (2.5%) diet containing high phosphorus (1.3% phosphorus and 0.7% calcium) and 0.3% adenine to induce uremia (SSNIFF Spezialdiäten GmbH). Control animals were fed same diet but without adenine. Every 10 days, a sample of blood was punctured at the tail vein of each animal. Serum and urine creatinine, calcium, and phosphorus were then assessed using standard colorimetric biochemistry analysis (Cobas Mira, Roche, Basel, Switzerland). When a fourfold increase of creatinine was observed compared to control, the rat was considered uremic and suitable for PK experiments. The method was validated using eight animals on control- and adenine-supplemented diet respectively. For PK studies, on the morning of the experimental day, a catheter was inserted in the right carotid artery under halothane anesthesia. Subsequently, rats were allowed to recover for 3 h and uremic and non-uremic rats, respectively, were assigned to three groups of six rats each. Balanced groups were obtained by assigning control animals with comparable body weight, a central PK-influencing parameter. The first group received 10 mg/kg $(OEG_2)_2$-IP4 injected i.v. via the catheter (1 mL/kg of a 10-mg/mL saline solution, pH 6.0), the second received s.c. 10 mg/kg $(OEG_2)_2$-IP4 (2 mL/kg of a 5-mg/mL saline solution, pH 6.0), and the third received 10 mg/kg IP6 injected i.v. via the catheter (1 mL/kg of a 10-mg/mL saline solution, pH 6.0). Blood aliquots were drawn after drug administration, centrifuged, decanted, and stored at $-80\,°C$ until serum drug quantification by LC-MS, as well as analysis of serum creatinine, calcium, and phosphorus (Cobas Mira).

**Efficacy of compounds in a vitamin D-induced VC model**. Male Sprague-Dawley rats were housed two per cage with free access to water and standard chow and submitted to a 12-h light/dark cycle. Compounds were administered and doses calculated using the salt form. Rats were randomly assigned to sham ($n = 7$), vehicle ($n = 11$), $(OEG_2)_2$-IP4 ($n = 11$), and IP6 ($n = 5$) groups and given a special diet containing low-protein (2.5%), high phosphorus (1.3%) and normal calcium (0.7%) (SSNIFF Spezialdiäten GmbH) for 12 days. Cardiovascular calcification was induced by s.c. injection (1 mL/kg) of 300,000 IU/kg/day of vitamin $D_3$ (vitamin $D_3$ Streuli, cholecalciferol, sol. inj. 300,000 ;U/mL, Streuli Pharma AG, Uznach, Switzerland) from day 1 to day 5. Sham groups were injected with 0.9% NaCl instead of vitamin $D_3$. Treatment was administered twice daily (second dose given 6 h after the first) starting from day 1 to day 12 by s.c. injection of $(OEG_2)_2$-IP4 (30 mg/kg twice a day in saline, pH = 6.0), IP6 (30 mg/kg twice a day in saline, pH = 6.0), or 0.9% NaCl (vehicle and sham group). Blood samples of 500 μL were collected on day 8 by fine needle from the tail vein and at sacrifice on day 12. After animals were euthanized by exsanguination under deep anesthesia (1–2% isoflurane, until decrease of the respiratory rate below 5 inspiration/min), the heart, and aorta were collected for subsequent analysis. Serum creatinine, calcium, and phosphorus were measured using standard colorimetric biochemistry analysis (Cobas Mira). Calcium tissue content was measured from dried carotids, whole left kidneys, and abdominal aortas by ICP-MS analysis. Qualitative staining for calcium by the von Kossa method, as well as hematoxylin and eosin histology staining were performed

on formalin-fixed dissected aortas, as described elsewhere[84]. Images of the rat aortic trees were acquired on a Canon 5D MARKIII (Canon, Tokio, Japan) equipped with a 50 mm lens.

In a subsequent experiment, animals were randomly assigned to six groups and given a special diet (SSNIFF Spezialdiäten GmbH) containing normal protein (17.6%), phosphorus (0.7%), calcium (1%), and warfarin (3 mg/g), starting from day 1 until day 6 of the study. The special diet did not contain vitamin K, therefore, a small amount of vitamin $K_1$ (1.5 mg/g) was added to prevent lethal bleeding. VC was induced by oral gavage (4 mL/kg) of 100,000 IU/kg/day of vitamin $D_3$ (D-CURE, cholecalciferol, 25,000 IU/mL, Leo Pharma, Ballerup, Denmark) on days 1, 2, 3, and 4. Treatment was administered by s.c. injections (5 mL/kg) of vehicle (saline, pH 7.4, $n = 40$), 12.5 mg/kg/day $(OEG_2)_2$-IP4 (saline, pH 7.4, $n = 12$), 25 mg/kg/day $(OEG_2)_2$-IP4 (saline, pH 7.4, $n = 12$), 50 mg/kg/day $(OEG_2)_2$-IP4 (saline, pH 7.4, $n = 19$), 25 mg/kg $(OEG_2)_2$-IP4 twice a day (saline, pH 7.4, $n = 18$), 50 mg/kg $(OEG_2)_2$-IP4 twice a day (saline, pH 7.4, $n = 19$), from day 1 to day 7. The first dose was given simultaneously with the daily vitamin D administration and the second dose (vehicle, if compound administration was only once a day), 6 h later. On day 8, animals were euthanized by exsanguination under deep anesthesia (intraperitoneal injection of ketamine 60 mg/kg and xylazin 7.5 mg/kg), blood and serum samples were collected, and the latter were stored at $-80\,°C$ for further analysis. At sacrifice, the heart, the aorta, carotids, and femoral arteries were harvested. Total calcium tissue content of the heart, the proximal part of the thoracic and abdominal aorta, and the left carotid and femoral arteries was measured by flame atomic absorption spectrometry, performed as described elsewhere[74], and calcium content was expressed as mg of calcium per g of dry tissue. Qualitative staining for calcium by the von Kossa method was performed on paraffin-embedded transversal sections of the thoracic and abdominal aorta at different positions along the vessel and were prepared as described previously[74] and images were taken on a Leica DMR microscope (Leica Camera AG) at ×50 magnification. Quantitation was performed by calculating the percentage of the calcified area on total tissue area using AxioVision image analysis software (Release 4.5, Carl Zeiss Microscopy GmbH) in which the total tissue area and the von Kossa-positive area were measured after applying two color separation thresholds. For each animal the average calcified area in percent was derived from calculating the percentage of calcified area of all thoracic and abdominal paraffin sections, respectively. For one animal each of group $1 \times 50$ mg/kg and $1 \times 12.5$ mg/kg, a reliable quantification of von Kossa staining of the a. thoracalis (Fig. 5h) and of bulk calcium in a. carotis (Fig. 5j), respectively, was not possible due to technical issues during tissue sampling and processing.

**Efficacy of compounds in an adenine and high-phosphate diet-induced VC rat model**. Male Wistar rats (Charles River) were housed two per cage with free access to water and standard chow and submitted to a 12-h light/dark cycle. Compounds were administered and doses calculated using the salt form. Animals were randomly assigned to four groups of 12 animals each and given a high-phosphate (1.03%) diet for 2 weeks before starting a special diet to initiate uremia. Uremia/VC was induced in all animals by freely fed diet containing high-phosphate (0.92%), low-protein (2.5% instead of 19%), and adenine (0.75%) for 4 weeks. The diet was again switched to a high-phosphate (1.03%) diet for 1 week, after which rats were sacrificed.

The first group was treated with vehicle (double distilled (d.d.) water), the second, third, and fourth groups were treated with daily doses equivalent of 5, 15, and 50 mg/kg of $(OEG_2)_2$-IP4, respectively, in 2 mL d.d. water starting 1 week after the administration of the adenine diet. Compounds were delivered continuously for 4 weeks via osmotic pumps (type 2ML4; Alzet, Cupertino, CA, USA) implanted subcutaneously under anesthesia 1 week after the start of the adenine diet. Blood sampling for serum and plasma at baseline, week 1, and week 3 was performed in restrained, conscious animals. Animals were sacrificed by exsanguination after anesthesia (i.p. administration of ketamine 60 mg/kg and xylazin 7.5 mg/kg) through the retro-orbital plexus. Serum samples were analyzed for creatinine according to the Jaffe method[85], calcium (using flame atomic absorption spectrometry[74]), and phosphate (DiaSys Diagnostic Systems, Holzheim, Germany) concentrations. In serum samples taken at sacrifice, the CPP-associated calcium was quantified by calculating the difference between total serum calcium (measured in serum supernatants after centrifugation at $3000 \times g$, RT, 15 min) and serum calcium after removing CPPs by centrifugation ($16,000 \times g$ and $4\,°C$ for 120 min)[26]. In addition, serum aliquots were centrifuged ($10,000 \times g$, $4\,°C$, 30 min) and $T_{50}$'s were determined as described above. CPP size was monitored by hourly DLS measurements at $37\,°C$, after spiking cleared serum samples with calcium (10 mM) and phosphate (6 mM), for a total of 8 h[29].

At sacrifice, the heart, the abdominal aorta, left carotids, and femoral arteries were collected. The presence of calcification in each animal was evaluated on paraffin-embedded, von Kossa stained sections of the thoracic aorta (as described previously[74]), as well as by measurement of the total calcium content in the heart and vascular tissue samples by flame atomic absorption spectrometry. Von Kossa-positive area was quantified (AxioVision Release 4.5 software) (vide supra).

**Sample size and treatment assignment**. Sample size was determined by logistical and resource considerations (potential for optimal animal housing and handling in the facility), as well as characteristics of the animal model (e.g., variation in

calcification degree) and mortality. The sample size was not based on statistical power considerations, as no prior data were available. Treatments (vehicle vs. compound treatment) were assigned randomly and both rats and label of treatment were randomized. Animals that were excluded from analysis are reported in the corresponding Results section. Although treatments were not blinded for investigators during the study period, all analyses of soft-tissue calcification were carried out in a blinded way.

**Ethical approval**. All experiments using animals comply with the relevant ethical regulations. The PK experiments were carried out in agreement with the Italian Law D.L.vo 4 marzo 2014, n. 26 and with authorization from the Ministry of Health, Italy (authorization number 433/2016-PR dated 4 April 2016). The experiments of the vitamin D (s.c.)-induced VC model were accepted by the institutional committee for the humane use of animals, Service de la consommation et des affaires vétérinaires du canton de Vaud, Lausanne, Switzerland (authorization number VD3178 dated 11 November 2016). All other animal experiments were carried out in agreement with the National Institutes of Health Guide for the Care and Use of Laboratory Animals 85-23 (1996) and were approved by the University of Antwerp Ethics Committee (2017–83 and 2018–21).

**Quantitation of human ionized serum calcium**. To assess reduction in ionized serum calcium, increasing concentrations of test compounds were incubated in human serum (Human AB serum, off-the-clot, HSER-ABPM; LuBioScience GmbH, Zurich, Switzerland) at RT and free ionized calcium levels were measured subsequently by the o-cresolphthalein method using the calcium colorimetric assay.

**Compounds characterization with NMR, high-resolution mass spectrometry, and high-performance liquid chromatography–ultraviolet absorbance (UV) detection**. Compounds were characterized by $^1$H-NMR and $^{31}$P-NMR, $^1$H COSY (400 MHz, D$_2$O; Bruker Avance III HD, Bruker BioSpin). Exact masses were determined by high-resolution mass spectrometry (HRMS) (electrospray ionization (ESI)-quadrupole-time-of-flight tandem mass spectrometry) (Bruker maXis; Bruker Daltonics, Billerica, MA, USA) in acetonitrile (ACN)/water (1:1 v/v). HRMS was calibrated using dedicated Tunemix ESI-TOF (positive polarity) (Agilent, Santa Clara, CA, USA) as reference mass list, the calibration spectrum +MS, 0.10–1.43 min, and enhanced quadratic calibration mode (only mass accuracy values with an error within ±3 ppm were considered). Purity of the optically active benzylated precursors, given the full conversion of the final hydrogenolysis step, was assessed by high-performance liquid chromatography (HPLC) 1260 Infinity with UV detection ($\lambda = 210$ nm) (Agilent Technologies, Santa Clara, CA, USA) using a Phenomenex Synergi Hydro-RP80A 250 × 4.6 mm, 4 µm (Phenomenex, Torrance, CA, USA). Compounds were dissolved at 1 mg/mL in MeOH, the injection volume was 5 µL, mobile phase A was Milli-Q water, and mobile phase B was ACN (+0.1% v/v formic acid). Gradient program was $t = 0$ min: A/B (70:30 v/v); $t = 10$ min: A/B (5:95 v/v); $t = 30$ min: A/B (5:95 v/v), and flow rate was 1 mL/min. Only compounds with >96% purity were considered for further experiments.

For the compound stress test, 20.25 mg (OEG$_2$)$_2$-IP4 in 700 µL D$_2$O was incubated at 37 °C and stability was monitored by $^1$H-NMR and $^{31}$P-NMR spectroscopies for up to 6 weeks.

**Compound quantitation by LC-MS/MS**. Plasma samples of (OEG$_2$)$_2$-IP4 and IP6 were analyzed on an ultra-fast liquid chromatography (UFLC) Shimadzu AC20 (Shimadzu) instrument coupled with an API 4500 Triple Quadrupole AB Sciex (Sciex, Danaher Corporation, Washington, DC, USA) mass spectrometer. For the LC part, mobile phase A was 10 mM NH$_4$HCO$_3$ in water (pH 8) for (OEG$_2$)$_2$-IP4 and 50 mM for IP6, mobile phase B was MeOH and injection volumes of 3 and 5 µL were used for (OEG$_2$)$_2$-IP4 and IP6, respectively. The column used for (OEG$_2$)$_2$-IP4 was Kinetex C8 50 × 2.1 mm, 2.6 µm 100 A (Phenomenex, Basel, Switzerland), thermostated at 35 °C. The column used for IP6 was Xterra MS C18 50 × 4.6 mm, 2.5 µm (Waters, Milford, MA, USA) thermostated at 40 °C. Different gradient programs were applied for (OEG$_2$)$_2$-IP4 (Supplementary Table 14) and IP6 (Supplementary Table 15) using a mobile phase flow rate of 0.3 mL/min. MS/MS parameters (multiple reaction monitoring (MRM) acquisition) for (OEG$_2$)$_2$-IP4 and IP6 together with internal standard (IS) are reported in Supplementary Tables 16 and 17, respectively. The source was working in electrospray negative ionization (ESI−) mode for both compounds with the following ESI parameters: for (OEG$_2$)$_2$-IP4, capillary temperature 400 °C, Gas1 50, Gas2 40, CUR 40, spray voltage −4500 V, CAD medium; for IP6, capillary temperature 350 °C, Gas1 30, Gas2 40, CUR 30, spray voltage −4500 V, CAD medium, DP −19, EP −8. Quantitation was carried out as total ion current on the sum of the MRM transitions. Calibration curve and quality control samples were prepared by adding 5 µL of working solution to 45 µL rat blank plasma and precipitating plasma protein by addition of 50 µL of 0.1% v/v trichloroacetic acid in water/MeOH (1:1 v/v) including IS (i.e., 1 µg/mL of OEG$_{11}$-IP5 for (OEG$_2$)$_2$-IP4, no IS for IP6). Samples were agitated 5 min on a 600 × $g$ orbital shaker and centrifuged (16,000 × $g$, 5 °C, 20 min). Supernatants (25 µL) were then transferred into a plate with 25 or 50 µL of 1% w/v NH$_4$OH in water for (OEG$_2$)$_2$-IP4 and IP6, respectively. Samples were then subjected to LC-MS/MS. Rat PK samples were prepared for analysis in an analogous manner as done for spiked samples. The lower limit of quantification (LLOQ)

and detection (LOD) in rat plasma for (OEG$_2$)$_2$-IP4 were 25 and 10 ng/mL, respectively. The upper limit of quantification (ULOQ) and detection was 5000 ng/mL. For IP6, LLOQ, LOD, and ULOQ were 500, 250, and 25,000 ng/mL, respectively.

The bioanalytical method for (OEG$_2$)$_2$-IP4 in rat plasma was further optimized, qualified, and applied to the comparative PK study in adenine and high-phosphate diet-induced VC vs. vitamin D-induced VC animals. The LC-MS/MS analyses were carried out using a Dionex Ultimate RSLC ultra-high-pressure liquid chromatography (UHPLC) system (Thermo Fisher Scientific) coupled with a TSQ Quantiva$^{TM}$ triple quadrupole mass spectrometer. The samples were stored at 5 °C in the autosampler prior to and during the analyses. Optimal separation was performed with a Waters (Milford, MA) XBridge BEH Amide, 2.1 × 50 mm, 2.5 µm maintained at 25 °C, and a flow rate of 0.4 mL/min. Gradient elution (solvent A, 20 mM ammonium acetate in water (pH 5); solvent B, ACN) was used according to the program reported in Supplementary Table 18. The injection volume was 10 µL. MS/MS parameters (MRM acquisition, cycle time 0.2 s) for (OEG$_2$)$_2$-IP4 are reported in Supplementary Table 19 and the ESI source, working in negative ionization mode, applied the following parameters: static spray voltage 3000 V ion transfer tube and vaporizer temperatures at 400 and 100 °C, respectively; sheath and auxiliary gas (nitrogen) flow rates at 80 and 3 (arbitrary units). For the quantitative determination in rat plasma, product ion $m/z$ 158.9 was selected because it provided the best sensitivity and selectivity. Calibration curve and quality control samples were prepared by adding 2 µL of working solution to 38 µL rat blank K$_3$EDTA plasma (Biochemed, Winchester, VA), and precipitating plasma protein by addition of 120 µL of MeOH including internal standard OEG$_{11}$-IP5. Samples were vortexed 1 min and centrifuged (22,000 × $g$, 4 °C, 10 min). Supernatants (90 µL) were then transferred into a glass vial with insert and injected onto LC-MS/MS system. Both in adenine and high-phosphate diet-induced VC and vitamin D-induced VC rat PK samples were prepared for analysis in an analogous manner as done for spiked samples (assay volume 40 µL). The LLOQ and LOD in rat plasma for (OEG$_2$)$_2$-IP4 were 50 and 10 ng/mL, respectively. The ULOQ and detection was 50,000 ng/mL.

**Calcium quantitation by inductivity coupled mass spectrometry (ICP-MS)**. Calcium tissues' contents were measured at the Central Environmental Laboratory (EPFL, Lausanne, Switzerland). Dry tissues (90 °C, overnight) were weighed and digested by aqua regia overnight at RT, followed by 2 h reflux, after which the solvent was completely evaporated with a steady flow of nitrogen gas. The dried samples were then reconstituted and homogenized with nitric acid. Calcium standards and samples were diluted with nitric acid. Absorbance was recorded using ICP-OES (Shimadzu ICPE-9000, Shimadzu). Two technical replicates were measured for each sample. Standard solutions were also included with the experimental samples for each set of measurements. Tissue concentrations of calcium were expressed as mg/g of dry tissue.

**Statistical analysis**. Data analysis, curve fitting, and statistical analysis were performed with Prism 8. Statistical significant differences between two groups were calculated using paired or unpaired Student's $t$-test. Statistically significant differences between multiple groups were derived by one-way ANOVA followed by Tukey's multiple comparison test. For in vivo data non-parametric testing by Kruskal–Wallis test followed by Mann–Whitney test with Bonferroni correction was performed. Normality of in vivo data was assessed using IBM SPSS 24. For the survival curve Kaplan–Meier analysis was performed and comparison of survival was done using a log-rank (Mantel–Cox) test. All data are presented as mean ± standard deviation (s.d.) from $n$ independent experiments, as indicated in each figure legend and a $p$ value of <0.05 was considered significant.

## Data availability

The authors declare that the data supporting the findings of this study are available within the paper and its Supplementary information files. The source data underlying Table 1, Figs. 2a, 2c–e, 3a, 3b, 3d–k 4a, 4c, 4e–l 5a–e, 5g–j, 6a, 6b–h and Supplementary Figs. 1–5, 10–13, 16, 18–23 and 26 are provided as a Source Data file. Reagents are available upon reasonable request from the corresponding authors (J.C.L. and M.I.).

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

## Acknowledgements

Financial support from Innosuisse (Grant 25174.1 PFLS-LS), technical support by the Scientific Center for Optical and Electron Microscopy (ScopeM), the Small Molecule Crystallography Center (SMoCC), and MS-Service LOC/ MoBiAS at the Swiss Federal Institute of Technology (ETHZ, Zurich, Switzerland), as well as support by the Central Environmental Laboratory (EPFL, Lausanne, Switzerland) is acknowledged. A.E.S. was supported by the scholarship fund of the Swiss Chemical Industry (SSCI). We thank S. Bisso for providing the PEG-alendronate control compound. B.C. was supported by the Natural Sciences and Engineering Research Council of Canada (NSERC) (discovery grant RGPIN-2015-05364).

## Author contributions

*Inositec*: M.E.I.—overall conception, synthesis, overall design, overall coordination, overall interpretation, manuscript writing, and revision. R.M.—design and coordination of in vivo studies, coordination of synthesis, manuscript revision. *ETH Zurich*: V.C.B.—execution of in vitro studies and manuscript revision. J.C.L.—overall conception, overall design, overall coordination, overall interpretation, manuscript writing and revision. A.E.S.—design, execution and analysis of all in vitro studies, leading manuscript writing and revision. *CHUV*: M.M., O.P., D.M., and M.B.—design and execution of vitamin D efficacy and adenine and high-phosphate diet-induced calcification PK studies, manuscript revision. D.S. and L.A.D.—bioanalytical method development and manuscript revision. *University of Antwerp*: P.C.D.—design of vitamin D/warfarin and adenine model efficacy studies and manuscript revision. E.N.—design of vitamin Dwarfarin and adenine model efficacy studies and manuscript revision. A.V.—design and execution of vitamin D/warfarin and adenine model efficacy studies, manuscript writing and revision. G.B.—analysis of bone μCT data and manuscript revision. *University of Edinburgh*: C.J.A.—design and execution of NaF-PET ex vivo studies and manuscript revision. M.G.M.—design and execution of NaF-PET CPP studies and manuscript revision. D.E.N.—design of NaF-PET CPP and ex vivo studies and manuscript revision. A.A.S.T.—design and execution of NaF-PET CPP and ex vivo studies and manuscript revision. *McGill University*: B.C.—design of compounds and synthesis, execution of selected synthesis, and manuscript revision. C.H.—synthesis of (OEG$_{12}$)-IP5 and (OEG$_7$)$_3$-IP5 and manuscript revision. A.L.—synthesis of (OEG$_{12}$)-IP5 and (OEG$_7$)$_3$-IP3, and manuscript revision.

## Competing interests

M.I., J.C.L., and B.C. are co-inventors of patents licensed to Inositec Inc. and shareholders thereof. Inositec's lead compound INS-3001 ((OEG$_2$)$_2$-IP4) is in development for the prevention and treatment of diseases driven by cardiovascular calcification. The remaining authors declare no competing interests.
