## [Peer Review File · Nature Communications]

Reviewers' comments:

Reviewer #1 (Remarks to the Author):

This study from a multidisciplinary author team identified an ethylene glycol-functionalized myo-inositol phosphate (OEG2)2-IP4 as an inhibitor of calcification on the chemical, cell culture and animal study level. The authors discuss that the compound hampers growth of vascular calcification directly by electrostatic interactions of (OEG2)2-IP4 at the level of nano-, micro- and macrocalcifications, while preserving calcium and phosphate homeostasis. Thus the compound is considered a potentially potent, safe, and universal inhibitor of pathological soft tissue calcification. This is significant given the fact that apart from surgical intervention hardly any therapy exists to treat existing soft tissue calcification and that preventive measures in dialysis patients are far from perfect.

The work includes synthesis and screening of compounds, optimization of a lead compound based on IP6 in vitro using a precipitation test, application of an optimized IP4 compound in a smooth muscle cell-based calcification test, as well as in animal experimentation employing two rat models of calcification. The manuscript presents a complete story comprising a 40-page main manuscript, 20 page online methods and 30 additional pages of supplementary information. Despite an overall good presentation, the sheer complexity of the work and the referral throughout the main manuscript to 31 supplementary figures and 9 supplementary tables makes reading hard. I wonder if all the information in the main manuscript is really necessary to convey the main message. At times the authors themselves got confused with referencing and table numbers. Also, the necessity to switch back and forth between figures and supplemental figures that address similar topics yet use different methods and modes of presentation (e.g. the stability experiments shown in Fig. 1f and Fig. S10) is confusing. I suggest to rearrange text and concatenate Figures (eg. Fig 1e with S10) to improve reading. I do like the completeness of the story and do not want to detract any of the information, but perhaps referring the entire material synthesis and characterization to the online supplement could help to shorten the main manuscript and improve reading. Nevertheless I commend the authors on a fine piece of work.

Intro

p3. the concentration product of calcium and phosphate in blood do NOT exceed the solubility product, they are metastable, i.e. they sustain crystal growth not nucleation (O'Neill WC. The fallacy of the calcium-phosphorus product. *Kidney Int.* 2007;72(7):792-796. doi:10.1038/sj.ki.5002412.)

p4. clearance of CPP2 indeed occurs in the MPS. CPP1 are however cleared first by endothelial cells und mediate what is called »phosphate toxicity« (Koeppert S, Büscher A, Babler A, Ghallab A, Buhl

EM, Latz E, Hengstler JG, Smith ER, Jahnen-Dechent W. Cellular Clearance and Biological Activity of Calciprotein Particles Depend on Their Maturation State and Crystallinity. *Front Immunol.* 2018;9:1991. doi:10.3389/fimmu.2018.01991.). This finding is important in the light of recent research stating that only CPP1 and even earlier CPP forms are physiological (Miura Y, Iwazu Y, Shiizaki K, Akimoto T, Kotani K, Kurabayashi M, Kurosu H, Kuro-o M. Identification and quantification of plasma calciprotein particles with distinct physical properties in patients with chronic kidney disease. *Sci Rep.* 2018;8(1):1-16. doi:10.1038/s41598-018-19677-4.), but CPP2 are probably confined to few cases and places like ascites in a deceased sclerosis peritonitis patient (Olde Loohuis KM, Jahnen-Dechent W, van Dorp W. The case: milky ascites is not always chylous. *Kidney Int.* 2010;77(1):77-78. doi:10.1038/ki.2009.407.) It is just so that CPP2 are easier made in the lab and are more stable and have therefor been employed in early cell and animal studies.

p8. CPP2 have sizes of up to 1 micrometer diameter. This is at variance with published values and indeed contradicts the TEM pictures in Fig 1 and Suppl fig 13. Please discuss.

p9. CPP binding strength was determined in a competitive assay against chemisorbed NaF. Does this mean that the (OEG₂)₂-IP₄ compound is also chemisorbed and if yes to what component of the CPP? Please discuss alternative competition mechanisms like diffusion barrier etc.

p10. The transdifferentiation of SMC into osteoblastic cells cannot be upheld in the presence of recent research and is indeed misleading. The »upregulation« of »osteogenic markers« is meaningless without proper osteoblast controls, and is not needed here. Please omit or discuss alternative mechanisms outlined in these papers.

1. Patel JJ, Bourne LE, Davies BK, Arnett TR, MacRae VE, Wheeler-Jones CP, Orriss IR. Differing calcification processes in cultured vascular smooth muscle cells and osteoblasts. *Exp Cell Res.* 2019;380(1):100-113. doi:10.1016/j.yexcr.2019.04.020.

2. Nguyen AT, Gomez D, Bell RD, Campbell JH, Clowes AW, Gabbiani G, Giachelli CM, Parmacek MS, Raines EW, Rusch NJ, Speer MY, Sturek M, Thyberg J, Towler DA, Weiser-Evans MC, Yan C, Miano JM, Owens GK. Smooth muscle cell plasticity: fact or fiction? *Circ Res.* 2013;112(1):17-22. doi:10.1161/CIRCRESAHA.112.281048.

3. Tang Z, Wang A, Yuan F, Yan Z, Liu B, Chu JS, Helms JA, Li S. Differentiation of multipotent vascular stem cells contributes to vascular diseases. *Nat Comms.* 2012;3(1):875. doi:10.1038/ncomms1867.)

p14 ff models of calcification. Instead of uremic and non-uremic model of calcification, please call the models what they are eg. VitD induced calcification and adenine induced calcification. VitD poisoning in rats causes general soft tissue calcification including arterial calcification, but also

nephrocalcinosis, myocardial and lung calcification. Did the compound affect any of the organ calcifications not mentioned in the paper?. Did the two VitD induced calcification models differ in outcome and why where they switched?

p17 li 2, I guess Supplementary table 7 in the text refers to Supplementary table 13 in the supplement. Please correct.

Fig 1e, please combine with Suppl Fig 9; asterisks and crosses mentioned in fig legend are missing in Figure. same in Suppl Fig 13.

Fig 2, please state all values in figures in mass units instead of percent of control to allow comparison with similar studies. same in Fig 2 I.

Fig 3. PK studies, were urine excretion of Ca and Pi measured? If yes please report.

Fig 5. heat calcification was measured in the adenine treated rats. Please report calcification in any other soft tissues found calcified during organ harvest i.e. kidney, lung

p29 Discussion of the pathological response of CPP2. Please include latest research showing that earlier forms of CPP and the lack of proper clearance and thus chronic exposure are more likely associated with calcification. (see lit mentioned regarding p4.) The way I see it the compound stabilizes CPP1, which can be more easily cleared unless it occurs in very high amounts and cannot be cleared at all. Therefore it is important to measure renal mineral clearance \pm compound (see comment for Fig 3)

p30 li 3 from bottom. Ref 21 states that »in Ahsg domain D1, surface binding of

calcium is mediated by negative charges on the extended β -sheet of domain D1 (top) that might occupy PO₄ positions on the (001) face of apatite crystals«. Do the authors contend that (OEG2)2-IP₄ compound likewise interacts with nascent apatitic calcium phosphate. Did they test if (OEG2)2-IP₄ compound alone stabilized supersaturated Ca and Pi solutions by forming colloids akin to CPP?

p31 ref 27 does not deal with the similarity of calcified lesions and bone, but reports the universal presence of spherical particles in calcified lesions, please omit or cite another reference (also see comment on p10)

p41 please check concentration of phosphates in phosphate stock solution. This adds up to 23 mM Pi altogether and does not result in 10 mM after dilution as stated a few lines below.

p42 phytase assay, typo »3-phyates«

p55 Ethical approval; why were the PK studies approved by an Italian Ministry when none of the authors is affiliated with an Italian institution?

References

please check that each reference occurs only once in the references list and vice versa. Ref 30 Irkle at all appears three times (also 47 and 70)

suppl Figure 11, the starting pH values are exceedingly high and incompatible with cell growth. Were these media gassed with CO₂? Also the pH differences are physiologically meaningful as they would affect cell and tissue health. Please discuss.

suppl Figure 14. Were CPP1 and CPP1 with (OEG2)2-IP4 compound also tested?

suppl Figure 18 please write out MRT

suppl Figure 22 please remove yellow background by proper white balance adjusting

suppl Figure 26 please compare baseline serum electrolytes in adenine model 1mM Pi ; 3.7 mM Ca with VitD model 3.5 Pi; 3.5 mM Ca, and sham animals 2 mM Pi; 2.7 mM Ca. Where these baseline differences expected? Please discuss.

signed

Willi Jahnen-Dechent

Reviewer #2 (Remarks to the Author):

This is a very interesting manuscript, where the authors have set out to improve the pharmacokinetic properties of an endogenous compound Myo-inositol hexakisphosphate (IP6) that inhibits vascular calcification using a synthetic chemistry strategy of analogues followed by a phenotypic screen. The authors have identified (OEG2)2-IP4 as the most promising candidate. They demonstrate in a very substantial and comprehensive series of experiments (including over 50 pages of supplementary information) (OEG2)2-IP4 has improved *in vitro* efficacy, as well as pharmacokinetic and safety profiles. The authors tested the compound *in vitro* in cultures of primary human vascular smooth muscle cells (VSMCs) and calcified human vascular tissue explants and suggest a mechanism of action *in vivo* activity was tested in acute and chronic, uremic and non-uremic rat models of vascular calcification. The authors are to be commended for these studies as proof of principle that (OEG2)2-IP4 functions *in vivo* and therefore strengthens the argument for translation into the clinic.

(1) Potency, specificity, *in vivo* compared to *in vitro* concentrations.

The authors note that (OEG2)2-IP4 is novel and some of the authors declare an interest that compounds have been patented. This is important for translation into the clinic. The authors should comment in the discussion.

The authors used a ¹⁸F-NaF-based competitive binding assay to indirectly measure affinity (IC₅₀) of the compounds in blocking ¹⁸F-NaF activity by chemisorption onto hydroxyapatite crystals of micro-calcification. The IC₅₀ of (OEG2)2-IP4 was calculated to be sub micro-molar, 0.81 μM. This was higher than IP6, 3.96 μM. Does (OEG2)2-IP4 bind reversibly in this assay?

In *in vitro* functional assays eg Figure 2, (OEG2)2-IP4 was tested over a wide concentration range, 0.01 – 100 μM. The EC₅₀'s by inspection from the graphs shown figures eg 2b and 2e varies to an extent, with effects in 2e at much higher concentrations, that might be expected for the IC₅₀ value above. Can the authors comment? Do the authors have any data on specificity ie have they tested (OEG2)2-IP4 against a panel of established drug targets in commercial screens that include receptors and ion channel? Or can the author comment perhaps as to why off-target effects are unlikely. Can the authors comment as to how these relate to values *in vivo*: C_{max} values obtained *in vivo* were ~50 μM following 10mg/kg *i.v* (Fig. 3) or higher values 50 mg/kg, p13), twice daily *sc* of 30 mg/kg, p25?

(2) Pharmacokinetics

Figure 3d is interesting as 10 mg/kg sc gave lower C_{max} values ~12 μM, AUC ~1200 versus 3B where 10 mg/kg iv gave AUC ~1600 and c_{max} ~50 μM. The sc route often acts as a depot to increase the concentration of drug over time. Was there any binding to plasma proteins with (OEG2)2-IP4? Did the authors calculate volumes of distribution as this may be informative as to whether (OEG2)2-IP4 is leaving the plasma compartment or remains bound.

The authors state p15, 'This model resulted in massive VC, which was observed primarily along the entire aorta (Supplementary Fig. 21) in the tunica media (Supplementary Fig. 22). (OEG2)2-IP4, dosed s.c. at 30 mg/kg twice daily for 11 days, significantly decreased formation of carotid VC as evidenced from calcium quantification by inductively coupled plasma mass spectrometry (Fig. 4a), but the decrease in abdominal aorta calcification was not significant (Supplementary Fig. 23). Could these differences be accounted for by variability in pharmacokinetics/physicochemical properties of (OEG2)2-IP4?

The authors tested in vitro stability of selected OEGylated inositol phosphates in enzymatically active human serum or after exposure to 3-phytase (an IP6-degrading enzyme). Importantly (OEG2)2-IP4 was active in vivo in rats but metabolism of compounds is often different to humans. Can the authors comment on any further data they may have on whether (OEG2)2-IP4 is likely to be effective in humans.

(3) Power calculations and blinding.

The authors state, p55. Methods Fig. 5. Study scheme of uremia-induced VC rat model.

5.4. Sample size and treatment assignment

Sample size was determined by logistical and resource considerations, as well as characteristics of the animal model (e.g., variation in calcification degree) and mortality. The sample size was not based on statistical power considerations, as no prior data were available. Treatments (vehicle vs. compound treatment) were assigned randomly. Animals that were excluded from analysis are reported in the corresponding results section. Blinding was not performed'

Authors are encouraged to consult the ARRIVE (Animal Research: Reporting of In Vivo Experiments

<https://www.nc3rs.org.uk/arrive-guidelines>) guidelines that lists key information to improve reporting of research on animals. The current information is brief and readers would appreciate more information about the study design and reasons for not blinding or carrying out power calculations.

(4) Experimental design and analysis (Fig 4)

The authors in the rat model of vascular calcification state the numbers in each group as follows:

‘vehicle (n = 10), groups treated with twice daily s.c. injections of 30 mg/kg (OEG2)2-IP4 (n = 11) or IP6 (n = 5), and sham (n = 7). One rat in the vehicle group died suddenly several hours after first vitamin D3 injection without explanation.’

In a study design where the group receiving the treatment/surgery to generate the model is expected to cause some mortality, it is usual to have more to compensate for this than in the sham control. However, the design is the other way round. Please could the authors clarify.

Reviewer #3 (Remarks to the Author):

The Design and Synthesis (page 7) of the inhibitors is not clearly described. The account of the chemistry is fragmented, partial, and confusing.

It would be far less confusing if the Authors were simply to present the Methods and Data for all twelve compounds, supplemented by NMR spectra to show compound identity and purity. At minimum, the authors should re-write the account to correct the errors (see below) and to explain clearly how and why the compounds were made, giving representative Methods and data for each class of compound.

Specific comments:

"For initial studies, OEG12-IP5... was synthesized in four steps starting from the protected 1,2,3,4,5-penta-O-benzoyl-myoinositol (Supplementary Fig. 1a)."

SI Fig 1a does not describe this synthesis, but that of OEG2-IP5. The synthesis does not start from the penta-O-benzoyl compound mentioned.

A synthesis of OEG12-IP5 is given in Methods page 36-37, although this does not start from the penta-O-benzoyl compound, either.

"It was purified... and characterized by [1H-NMR] Supplementary Fig. 2 and 3), [31P NMR] (Supplementary Fig. 4 and 5)..."

These figures (pages 68 and 69) give NMR spectra for a completely different compound (OEG2)2-IP4, which is commercially available. No methods for the synthesis of (OEG2)-IP4 = INS-3001, the lead compound which features in most of the biology, are given here.

"... as well as [HPLC-UV] (Supplementary Fig. 6)..."

This figure (page 70) gives HPLC data (I think!) for the precursor to (OEG2)2-IP4 (compound 9), not for OEG12-IP5, nor for (OEG2)2-IP4 itself, presumably because these compounds lack good chromophores for detection by UV, although this is not made clear. 1H and 31P NMR spectra for all compounds tested would be better, direct, evidence of purity.

"In total, twelve compounds... were produced...and details on their synthesis and characterization can be found in the Supplementary Materials and Methods."

The Methods give details for the synthesis of only *two* compounds [(OEG)12-IP5) and (OEG7)3-ip3]. These details are well described, but there are no such details for the other ten compounds. The Supplementary gives Schemes (but no Methods or data) for three compounds, plus NMR spectra [wrongly attributed to (OEG)12-IP5 in the text] for one of these.

Corrections:

Inositol hexakisphosphate, pentakisphosphate and inositol, tetrakisphosphate throughout. Not "hexaphosphate", "pentaphosphate" and "tetrphosphate".

Page 70, legend: "...optically active benzylated precursor..." It is not correct to describe this compound as "optically active" which normally refers to the ability of a compound to rotate the plane of polarized light. All the compounds are either meso-compounds or racemic.

Supplementary table 7, page 90: mass spectrometric detection of multiply charged ions (z = 2+, z = 3+) is the calculation of m/z correct?

Query:

Page 7, Design of compounds: the chelator previously synthesised by this group (ref 42) had the PEG attached to the O2-atom of myo-inositol pentakisphosphate. This is logical because this manner of functionalization gives a meso-compound. The inositol pentakisphosphate derivatives in the present work have OEG attached to the 4/6-O atom, making each compound a racemic mixture, containing two enantiomers with conceivably different biological activities/toxicities. Why did the authors choose this strategy?

Reviewer #4 (Remarks to the Author):

This is a highly interesting study, extending the concept of using IP6 as treatment strategy for vascular calcification – presumably as a n inhibitor of calcium-phosphate crystallization. The derivate of IP6 investigated here shows potent inhibitory action on vascular calcification. The concept is highly interesting, and the authors employ some very interesting methodology to characterize their compound, although there is shortage on a deeper characterization of the models. Also the proposed mechanism is not presented in a clear and conclusive manner. Thus, I have noted some points for the authors consideration, which may improve the manuscript.

- The authors consider the investigated compound as treatment option. However, clearly the bone mineralization may be affected by the mode of action. The effects of the compound on the bone is not sufficiently investigated, and this limitation not sufficiently discussed, since similar to the adenin model, CKD patients may be the prime patient population for treatment. The bone structure should be documented by μ CT. Is there an effect of the compound on the mechanical properties? Am I missing the effects of the compound on bone in healthy animals – this would seem decisive?

- The effects in the animal models should also be characterized better, especially in the uremic model: what are the concentrations of PTH and FGF23? The beneficial effect on renal function in the Vitamin D model imply, that there may be calcification in the renal system, which causes loss of renal function. What are the effects of treatment on renal calcium content (analogous to aortic calcium content)

- No clear mechanism of action is described, but some suggestion of a Fetuin-like effect to prevent formation of secCPPs. secCPPs increase intracellular calcium levels, induces oxidative stress and

inflammatory responses, as already described mechanisms required for inducing VSMCs calcification. Do the compounds inhibit these changes? How (OEG2)2-IP4 suppresses VSMC calcification? Further experiments clarifying at what point (OEG2)2-IP4 interferes with the calcification process in vitro would be beneficial.

Also, there are then discrepancies in the mechanism of action presented by the authors. On the one hand, based on serum calcification propensity data, the authors suggest that (OEG2)2-IP4 inhibits formation of secCPPs and thus, presumably, prevents vascular calcification. However, in vivo, the T50 of serum from (OEG2)2-IP4 treated animals was not different than from vehicle animals. On the other hand, authors show that (OEG2)2-IP4 inhibits VSMCs calcification in vitro induced by already formed secCPPs.

- Along these lines: The effects on VSMC osteoprotegerin expression is surprising. What are the expression levels of vsmcs and calcification markers, such as alpha smooth muscle actin, RUNX2, BMP2, ALPL or Tnf-alpha? What is their expression in vascular tissue of the animals? How do authors explain no effects of secCPPs on ALPL activity?

- Fig. 2g. Cytokine levels of TNFa should also be determined in the supernatant of VSMCs. secCPPs-induced VSMC calcification involves increased levels of TNFa and activation TNFa/TNFR1 system in VSMCs.

- The comparative effects and concentration dose responses of the compounds on T50 in conditions without serum should be performed, to determine whether the compounds inhibit directly the precipitation of calcium and phosphate and formation of hydroxyapatite.

- Fig. 4.d-j. Control group without calcification is missing.

- Overall, n=3 is a low n number for the in vitro experiments.

- In figure 1d, why did the authors use 100 µM concentrations of the compounds, while in previous data much lower concentrations were used? The control group without addition of compounds is missing.

Minor comments

-The introduction is rather unspecific and superficial when distinguishing between medial, intimal calcification and even valvular calcification. There are some profound differences in these processes. This should be improved. This explains also why the authors used VSMCs and macrophages. This also extends to the discussion. Also, there is yet no evidence that reducing calcification is of benefit to mortality.

- Sometimes is written mouse model, p15 last sentence – the methods and figure legends do not clearly state then which animal species and which background was used

- Methods 2.3 is too superficial; calcium and phosphate solutions need to be specified: concentration and what kind of phosphate/calcium was used (or add as described above in 2.1).

- Injections in the vitamin D-Warfarin model: “once or twice daily” appears rather random?

Reviewer #1

Comment 1: Despite an overall good presentation, the sheer complexity of the work and the referral throughout the main manuscript to 31 supplementary figures and 9 supplementary tables makes reading hard. I wonder if all the information in the main manuscript is really necessary to convey the main message. At times the authors themselves got confused with referencing and table numbers. Also, the necessity to switch back and forth between figures and supplemental figures that address similar topics yet use different methods and modes of presentation (e.g. the stability experiments shown in Fig. 1f and Fig. S10) is confusing. I suggest to rearrange text and concatenate Figures (eg. Fig 1e with S10) to improve reading. I do like the completeness of the story and do not want to detract any of the information, but perhaps referring the entire material synthesis and characterization to the online supplement could help to shorten the main manuscript and improve reading.

Our response:

The introduction and conclusion were amended and condensed in response to multiple remarks, and figures were combined or removed to improve reading (e.g., Fig 1 now includes former Supplementary Fig. 9 and 10, former Supplementary Fig. 17 (ALP activity) was removed). Moreover, we restructured the online supplementary to reduce complexity (e.g., by including a title page with index, and by consolidating all information related to compound synthesis and characterization, including Supplementary Figures 1-6 into a chapter called “3. Supplemental Methods, 3.1. Chemistry”).

Comment 2: p3. the concentration product of calcium and phosphate in blood do NOT exceed the solubility product, they are metastable, i.e. they sustain crystal growth not nucleation (O'Neill WC. The fallacy of the calcium-phosphorus product. *Kidney Int.* 2007;72(7):792-796. doi:10.1038/sj.ki.5002412.)

Our response:

We took notice of the reference and corrected the sentence accordingly (p3, line 55).

Comment 3: p4. clearance of CPP2 indeed occurs in the MPS. CPP1 are however cleared first by endothelial cells und mediate what is called »phosphate toxicity« (Koeppert S, Büscher A, Babler A, Ghallab A, Buhl EM, Latz E, Hengstler JG, Smith ER, Jahnen-Dechent W. Cellular Clearance and Biological Activity of Calciprotein Particles Depend on Their Maturation State and Crystallinity. *Front Immunol.* 2018;9:1991. doi:10.3389/fimmu.2018.01991.). This finding is important in the light of recent research stating that only CPP1 and even earlier CPP forms are physiological (Miura Y, Iwazu Y, Shiizaki K, Akimoto T, Kotani K, Kurabayashi M, Kurosu H, Kuro-o M. Identification and quantification of plasma calciprotein particles with distinct physical properties in patients with chronic kidney disease. *Sci Rep.* 2018;8(1):1-16. doi:10.1038/s41598-018-19677-4.), but CPP2 are probably confined to few cases and places like ascites in a deceased sclerosis peritonitis patient (Olde Loohuis KM, Jahnen-Dechent W, van Dorp W. The case: milky ascites is not always chylous. *Kidney Int.* 2010;77(1):77-78. doi:10.1038/ki.2009.407.) It is just so that CPP2 are easier made in the lab and are more stable and have therefor been employed in early cell and animal studies.

Our response:

We rephrased the passage on CPP clearance in the introduction taking into consideration all and citing some of the above mentioned literature (p4, line 74).

Reviewer #1 Comment 4: p8. CPP2 have sizes of up to 1 micrometer diameter. This is at variance with published values and indeed contradicts the TEM pictures in Fig 1 and Suppl fig 13. Please discuss.

Our response:

The CPP size we obtained by TEM is in line with published values, whereas CPP size obtained by dynamic light scattering (DLS) was slightly larger. Particle size information from TEM images often do not corroborate well with data obtained from DLS. DLS is a technique based on the fluctuation of the intensity of scattered light and deduces the hydrodynamic radius from Brownian motion of dispersed particles (dynamic state). It can often not discriminate individual particles from aggregates. TEM is a number-based technique applied to dry samples under ultrahigh vacuum conditions (static state). Hence, the size obtained by DLS is usually larger than TEM¹. We now briefly discuss this in the manuscript (p. 8, line 176).

Reviewer #1 Comment 5: p9. CPP binding strength was determined in a competitive assay against chemisorbed NaF. Does this mean that the (OEG₂)₂-IP4 compound is also chemisorbed and if yes to what component of the CPP?

Our response:

Yes, we believe (OEG₂)₂-IP4 is chemisorbed to the hydroxyapatite core of CPPs. We have now cited this in the manuscript (p. 9, line 194). The molecule is small (ca. 16 Å in length) and therefore likely not retained by the protein layer covering CPPs. Similarly, alendronate-PEG conjugates have been shown to bind to bone, despite the fact that bony tissue is lined by the endosteum or periosteum².

Please discuss alternative competition mechanisms like diffusion barrier etc.

Our response:

We believe there are no other competition mechanisms. It is unclear to us what is meant with “diffusion barrier”. Should this refer to the protein corona of CPP2s, we again believe that this should not hinder (OEG₂)₂-IP4 from interacting with the HAP core, due to the molecule’s relatively small size (*vide supra*).

Reviewer #1 Comment 6: p10. The transdifferentiation of SMC into osteoblastic cells cannot be upheld in the presence of recent research and is indeed misleading. The »upregulation« of »osteogenic markers« is meaningless without proper osteoblast controls, and is not needed here. Please omit or discuss alternative mechanisms outlined in these papers.

1. Patel JJ, Bourne LE, Davies BK, Arnett TR, MacRae VE, Wheeler-Jones CP, Orriss IR. Differing calcification processes in cultured vascular smooth muscle cells and osteoblasts. *Exp Cell Res.* 2019;380(1):100-113. doi:10.1016/j.yexcr.2019.04.020.

2. Nguyen AT, Gomez D, Bell RD, Campbell JH, Clowes AW, Gabbiani G, Giachelli CM, Parmacek MS, Raines EW, Rusch NJ, Speer MY, Sturek M, Thyberg J, Towler DA, Weiser-Evans MC, Yan C, Miano JM, Owens GK. Smooth muscle cell plasticity: fact or fiction? *Circ Res.* 2013;112(1):17-22. doi:10.1161/CIRCRESAHA.112.281048.

3. Tang Z, Wang A, Yuan F, Yan Z, Liu B, Chu JS, Helms JA, Li S. Differentiation of multipotent vascular stem cells contributes to vascular diseases. Nat Comms. 2012;3(1):875. doi:10.1038/ncomms1867.)

Our response:

As suggested, Fig. 1h and Fig. 1i and the corresponding text in the main part were removed from the revised manuscript. Concurrently, we have also omitted Supplementary Figure 17 (ALP activity).

Reviewer #1 Comment 7a: p14 ff models of calcification. Instead of uremic and non-uremic model of calcification, please call the models what they are eg. VitD induced calcification and adenine induced calcification.

Our response:

As suggested, instead of uremic and non-uremic model of calcification, we now wrote vitamin D-induced calcification and adenine and high-phosphate diet-induced calcification model (changes were made throughout the entire work and highlighted in yellow).

Reviewer #1 Comment 7b: VitD poisoning in rats causes general soft tissue calcification including arterial calcification, but also nephrocalcinosis, myocardial and lung calcification. Did the compound affect any of the organ calcifications not mentioned in the paper?

Our response:

In the first vitamin D-induced VC rat study, vessels and kidneys were examined for calcification in the treated animals, and we now provide data on kidney calcium content in Fig. 4b of the revised manuscript (p. 26). In the second vitamin D-induced VC rat study, calcification was also assessed in the heart and the left kidney. In these organs the calcium content was negligibly low (< 0.2 mg/g tissue and <0.5 mg/g tissue in the heart and kidney, respectively) and no differences could be observed between treatment groups. We now state this in the revised manuscript (p. 16, line 379).

There is indeed published literature on vitamin D-induced VC models that report calcification in other organs (e.g., stomach, trachea and lung³), however, different vitamin D doses and dosing schedules and the source of the employed vitamin D appear to lead to variability in the manifestation of the calcification.

Reviewer #1 Comment 7c: Did the two VitD induced calcification models differ in outcome and why where they switched?

Our response:

The first vitamin D-induced VC rat model was much more severe and erratic in outcome (cf. i) bulk calcium and ii) body weight), and importantly the model also severely impacted animal behavior, since food intake and body weight decreased. Hence, we sought to improve the model so that calcification was induced in a steadier (slower) and less erratic (more robust) way (100 000 IU/kg/day Vitamin D p.o. for 4 days and a diet supplemented with warfarin and vitamin K1 (to reduce risk of bleeding), compared to s.c. injection of 300 000 IU/kg/day of vitamin D for 5 days). By doing so, animal suffering was also significantly reduced. We mentioned this in the manuscript (p. 16, line 357)

- i) Bulk calcium in the carotis of vehicle treated animals was 69 ± 47 mg calcium/g tissue (n = 10) in the first study compared to 26 ± 22 mg calcium/g tissue (n = 25) in the second study
- ii) Body weight dropped by 20% in vitamin D treated animals compared to shams in the first study (Supplementary Fig. 13), whereas body weight did not drop significantly in vitamin D-treated animals on day 7 compared to day 0 in the second study (Supplementary Fig. 18)

Reviewer #1 Comment 8: p17 li 2, I guess Supplementary table 7 in the text refers to Supplementary table 13 in the supplement. Please correct.

Our response:

This mistake was corrected (p. 17, line 396)

Reviewer #1 Comment 9: Fig 1e, please combine with Suppl Fig 9;

Our response:

We combined Supplementary Fig. 9 with Figure 1e (p. 20, Figure 1e).

asterisks and crosses mentioned in fig legend are missing in Figure. same in Suppl Fig 13.

Our response:

The asterisks and crosses were introduced to mark matching calculated d-spacings from SAED images of CPPs+IP6 and CPP2s respectively. The SAED images in Figure 1d are too small to properly denote d-spaces. This has also not been done in other reports of SAED images of CPP2s⁴. To improve reading, we have moved the paragraph on d-spacings from the figure legend to the main text (p. 9, line 184).

Reviewer #1 Comment 10: Fig 2, please state all values in figures in mass units instead of percent of control to allow comparison with similar studies. same in Fig 2 I.

Our response:

Since we observed different mass units of calcium deposition in the controls of independent assays, we decided to present the data as % of control instead of mass units. However, we have now included the range of calcium deposition in the control group in mass units in the legend of Figure 2 to allow comparison with similar studies (p. 22, Fig. 2b, e).

Reviewer #1 Comment 11: Fig 3. PK studies, were urine excretion of Ca and Pi measured? If yes please report.

Our response:

They were not measured in neither the PK, nor the efficacy studies.

Reviewer #1 Comment 12: Fig 5. heat calcification was measured in the adenine treated rats. Please report calcification in any other soft tissues found calcified during organ harvest i.e. kidney, lung

Our response:

As at sacrifice, only the heart, the aorta, carotids and femoral arteries were taken, no data on calcification in organs other than these are available from the present study.

Reviewer #1 Comment 13: p29 Discussion of the pathological response of CPP2. Please include latest research showing that earlier forms of CPP and the lack of proper clearance and thus chronic exposure are more likely associated with calcification. (see lit mentioned regarding p4.)

Our response:

We have included the latest research on CPPs and the respective literature now in the introduction (p. 4, line 74 ff.).

The way I see it the compound stabilizes CPP1, which can be more easily cleared unless it occurs in very high amounts and cannot be cleared at all. Therefore it is important to measure renal mineral clearance ± compound (see comment for Fig 3)

Our response:

We have amended the discussion in the revised manuscript with respect to the hypothesis on the mechanism of action of (OEG₂)₂-IP4 postulated by the reviewer. (p. 32, line 614 ff.). We have further addressed the mechanism of action in response to Reviewer #4 Comment 3.

Reviewer #1 Comment 14: p30 li 3 from bottom. Ref 21 states that »in Ahsg domain D1, surface binding of calcium is mediated by negative charges on the extended beta-sheet of domain D1 (top) that might occupy PO4 positions on the (001) face of apatite crystals«. Do the authors contend that (OEG₂)₂-IP4 compound likewise interacts with nascent apatitic calcium phosphate. Did they test if (OEG₂)₂-IP4 compound alone stabilized supersaturated Ca and Pi solutions by forming colloids akin to CPP?

Our response:

As suggested, we assessed whether the compounds inhibit directly the precipitation of calcium and phosphate and formation of hydroxyapatite with the assay from Heiss, A. *et al.*⁵. In brief, compound, 10 mM calcium solution and 6 mM phosphate solution in Hepes buffered saline (pH 7.4, RT) were incubated for 90 min at 37 °C and centrifuged at 8 000 x g for 5 min. The recovered pellet was dissolved in 1% (v/v) acetic acid and the calcium content quantified calorimetrically. (OEG₂)₂-IP4 alone stabilized supersaturated calcium and phosphate in solution dose dependently, as opposed to IP6 which was barely effective (Rebuttal Letter Figure 1). At 1 mM (OEG₂)₂-IP4 no precipitate formed and particles with sizes in the range of CPP1s (R_h < 100 nm) were detected by dynamic light scattering. These results suggest a synergistic effect of (OEG₂)₂-IP4 with endogenous calcification inhibitors, such as fetuin-A in serum. However, since the effect was observed at a rather high concentration under conditions that are not physiological (no serum), we prefer not to include these additional data in the manuscript.

Rebuttal Letter Figure 1 Inhibition of calcium phosphate precipitation in a serum-free environment. Mean \pm s.d. (n = 3).

Reviewer #1 Comment 15: p31 ref 27 does not deal with the similarity of calcified lesions and bone, but reports the universal presence of spherical particles in calcified lesions, please omit or cite another reference (also see comment on p10)

Our response:

We have removed this reference in the revised manuscript.

Reviewer #1 Comment 16: p41 please check concentration of phosphates in phosphate stock solution. This adds up to 23 mM Pi altogether and does not result in 10 mM after dilution as stated a few lines below.

Our response:

We have rephrased the paragraph to prevent misunderstandings (p. 40, line 775). The phosphate solution contained in total 24.00 mM Pi resulting from the combination of 19.44 mM Pi from Na₂HPO₄ and 4.56 mM Pi from NaH₂PO₄ * 2 H₂O. After 4-fold dilution into the assay plate, the final concentration was 6.00 mM Pi.

Reviewer #1 Comment 17: p42 phytase assay, typo »3-phyates«

Our response:

The typo was corrected (p. 41, line 795).

Reviewer #1 Comment 18: p55 Ethical approval; why were the PK studies approved by an Italian Ministry when none of the authors is affiliated with an Italian institution?

Our response:

The PK studies were performed by a contract research organization located in Italy. We have now included this information in the Methods section 5.1. Pharmacokinetics (p. 47, line 973)

References

Reviewer #1 Comment 19: please check that each reference occurs only once in the references list and vice versa. Ref 30 Irkle at all appears three times (also 47 and 70)

Our response:

We made sure that no reference occurs more than once.

Reviewer #1 Comment 20: suppl Figure 11, the starting pH values are exceedingly high and incompatible with cell growth. Were these media gassed with CO₂?

Our response:

Yes, these media were incubated in a humidified atmosphere with 5% CO₂.

Also the pH differences are physiologically meaningful as they would affect cell and tissue health. Please discuss.

Our response:

The difference in the starting pH of these media are not statistically significant and therefore the assays are still comparable. The observed difference may have resulted from a delay in pH measurement on the laboratory bench and hence a degassing of CO₂ and subsequent rise in pH.

Reviewer #1 Comment 21: suppl Figure 14. Were CPP1 and CPP1 with (OEG₂)₂-IP4 compound also tested?

Our response:

We have performed the requested *in vitro* experiment. The results can be found in Fig. 2g (p. 22). Incubating VSMCs with CPP1 increased calcium deposition compared to control, but the increase was not statistically significant and (OEG₂)₂-IP4 treatment starting at 10 μM reduced calcium deposition to the level of control. CPP1 are very delicate to handle and destabilized quite rapidly. We therefore always prepared fresh batches immediately before starting the assay, and a different batch was used in each independent assay.

Reviewer #1 Comment 22: suppl Figure 18 please write out MRT

Our response:

Corrected as suggested (Supplementary information p. 12, Supplementary Fig. 11).

Reviewer #1 Comment 23: suppl Figure 22 please remove yellow background by proper white balance adjusting

Our response:

We corrected this and removed the yellow background from the respective image, which is now Supplementary Fig. 15.

Reviewer #1 Comment 24: suppl Figure 26 please compare baseline serum electrolytes in adenine model 1mM Pi ; 3.7 mM Ca with VitD model 3.5 Pi; 3.5 mM Ca, and sham animals 2 mM Pi; 2.7 mM Ca. Where these baseline differences expected? Please discuss.

Our response:

A mistake was made when converting units of serum phosphate concentrations. After correction, serum phosphate in the vehicle animals at baseline now are identical in the vitamin D and adenine models (3.5 mM). Supplementary Fig. 20 (former Supplementary Fig. 26) was adapted accordingly in the revised manuscript. The statistically significant difference in baseline calcium levels observed in these animals may be explained as listed below (Supplementary Information, p. 17).

- i) The rats of the adenine-induced and vitamin D induced VC model, respectively, received different diets. The low-protein, adenine-supplemented diet is a synthetic diet with Ca derived from inorganic sources. It is well known that Ca from inorganic sources is much more bioavailable compared to Ca from organic sources.
- ii) Electrolytes were measured in plasma and serum, respectively.
- iii) Different rat strains were used: Sprague-Dawley rats in the vitamin D model and Wistar rats in the adenine model

Reviewer #2

(1) Potency, specificity, *iv vivo* compared to *in vitro* concentrations.

Reviewer #2 Comment1: The authors note that (OEG₂)₂-IP4 is novel and some of the authors declare an interest that compounds have been patented. This is important for translation into the clinic. The authors should comment in the discussion.

Our response:

We have now cited all relevant patents in the revised manuscript (p. 34, line 668). Preclinical studies with the lead compound have been initiated by Inositec in view of subsequent clinical development and while this is indeed important, we feel that it is not appropriate for the discussion of an article in a science-driven peer-reviewed journal. Given that certain authors disclose in the Competing Interests section that they are shareholders of Inositec and that INS-3001 ((OEG₂)₂-IP4) is in development by Inositec Inc., we feel that the provided information is sufficient for the interested reader. More information on the development of the compound as a drug candidate can be retrieved from the company's website.

Reviewer #2 Comment2: The authors used a 18F-NaF-based competitive binding assay to indirectly measure affinity (IC₅₀) of the compounds in blocking 18F-NaF activity by chemisorption onto hydroxyapatite crystals of micro-calcification. The IC₅₀ of (OEG₂)₂-IP4 was calculated to be sub micro-molar, 0.81 μM. This was higher than IP6, 3.96 μM. Does (OEG₂)₂-IP4 bind reversibly in this assay?

Our response:

Due to the electrostatic nature of the interaction of (OEG₂)₂-IP4 with hydroxyapatite we can speculate that the binding is reversible. For instance, the bisphosphonate alendronate binds rapidly and reversibly to human bone particles with a dissociation constant of ca. 1 mM⁶. Indeed, this assay was conducted at a single incubation time point and temperature optimized for NaF, in order to rank and compare (OEG₂)₂-IP4 relative to IP6 rather than investigate reversibility of the compounds' binding, which would require multiple incubation time points and/or temperature. However, assessing reversibility of binding is something we would consider in future studies. We addressed this in the revised manuscript (p. 9, line 194).

Reviewer #2 Comment3: In *in vitro* functional assays eg Figure 2, (OEG₂)₂-IP4 was tested over a wide concentration range, 0.01 – 100 μM. The EC₅₀'s by inspection from the graphs shown figures eg 2b and 2e varies to an extent, with effects in 2e at much higher concentrations, that might be expected for the IC₅₀ value above. Can the authors comment?

Our response:

We show good correlation of *in vitro* and cell culture results, as evident from comparing concentration ranges used in the screening assay (0.1 – 100 μM Fig. 1c) and the cell culture experiments in Fig. 2b (0.01 – 100 μM) and 2e (1 – 100 μM). EC₅₀s for these three experiments were all in the low μM-range. For the CaP-induced VSMC calcification assay (Fig. 2b) indeed ca. 10-fold higher concentration were necessary to achieve same activity as in the CPP-induced VSMC calcification assay (Fig. 2e), which can be explained by the different

mechanism of action on how calcification is obtained and prevented by (OEG₂)₂-IP₄ in these models. In the CaP assay low inhibitor concentration are sufficient to poison crystal nucleation, whereas for CPP assay higher inhibitor concentrations are needed to achieve sufficient particle stabilization by chemisorption of inhibitor onto CPPs. Indeed, this has been observed already previously with hydrogen sulfide. Hydrogen sulfide significantly prevented high phosphate medium-induced and CPP2-induced VSMC calcification at 50 μM and 300 μM, respectively^{7,8}. This was discussed in the revised version of the manuscript (p. 12, line 259)

Reviewer #2 Comment3b: Do the authors have any data on specificity ie have they tested (OEG₂)₂-IP₄ against a panel of established drug targets in commercial screens that include receptors and ion channel? Or can the author comment perhaps as to why off-target effects are unlikely.

Our response:

New results from off-target screening assays were included in the manuscript (p. 15, line 334, Supplementary Information, p. 27, Supplementary Table 11 and Supplemental Methods, p. 35)

Reviewer #2 Comment3c: Can the authors comment as to how these relate to values in vivo: C_{max} values obtained in vivo were ~50 μM following 10mg/kg i.v (Fig. 3) or higher values 50 mg/kg, p13), twice daily sc of 30 mg/kg, p25?

Our response:

For many other drugs, it is often difficult to correlate the C_{max} with the pharmacological activity, as the C_{max} may largely exceed the IC₅₀ (4.9 μM and 35.0 μM for (OEG₂)₂-IP₄ and IP₆, respectively). What is more important is to maintain the plasmatic levels of the compound above the IC₅₀ for a sufficient duration. Our pharmacokinetic data show that after s.c. injection plasmatic concentrations above the IC₅₀ were achieved for at least 4 h vs. 12 min for IP₆ (Fig. 3c), which partly explain the higher efficacy achieved with (OEG₂)₂-IP₄. We added this into the discussion of the revised manuscript (p. 31, line 589).

Reviewer #2 Comment4: Figure 3d is interesting as 10 mg/kg sc gave lower C_{max} values ~12 μM, AUC ~1200 versus 3B where 10 mg/kg iv gave AUC ~1600 and cMax ~50 μM. The sc route often acts as a depot to increase the concentration of drug over time. Was there any binding to plasma proteins with (OEG₂)₂-IP₄?

Our response:

We have further carried out *in vitro* experiments to assess protein binding of (OEG₂)₂-IP₄ in human plasma. The protein bound fraction of (OEG₂)₂-IP₄ in healthy human plasma was 58 and 42% calculated over total measured concentrations of 2.8 and 27.5 μg/mL (i.e., 4.0 and 39.5 μM, respectively), suggesting a potential saturation of the binding sites at high concentrations. These new data have been included in the revised manuscript (p 14., line 320 and Supplemental Methods 3.3, p. 35, line 446)

Did the authors calculate volumes of distribution as this may be informative as to whether (OEG₂)₂-IP₄ is leaving the plasma compartment or remains bound.

Our response:

(OEG₂)₂-IP4 showed a very low volume of distribution (234 mL/kg) which is expected for a hydrophilic molecule. The volume of distribution was added to the revised manuscript (p. 13, line 292).

Reviewer #2 Comment5: The authors state p15, ‘This model resulted in massive VC, which was observed primarily along the entire aorta (Supplementary Fig. 21) in the tunica media (Supplementary Fig. 22). (OEG₂)₂-IP4, dosed s.c. at 30 mg/kg twice daily for 11 days, significantly decreased formation of carotid VC as evidenced from calcium quantification by inductively coupled plasma mass spectrometry (Fig. 4a), but the decrease in abdominal aorta calcification was not significant (Supplementary Fig. 23). Could these differences be accounted for by variability in pharmacokinetics/physicochemical properties of (OEG₂)₂-IP4?

Our response:

It is unlikely that the observed differences result from a variability in the physicochemical properties of (OEG₂)₂-IP4 because the compound was sufficiently well characterized with respect to chemical identity. We believe that the different efficacy in reducing calcification in the carotid and abdominal aorta primarily results from the high heterogeneity observed in the response to the vitamin D treatment (i.e. calcification) within the vehicle and (OEG₂)₂-IP4 groups (hence the large number of rats that had to be sacrificed). A trend towards decreased formation of calcification in the abdominal aorta in the compound treated group was observed but not found to be statistically significant (Supplementary Information, p. 15, Supplementary Fig. 16).

Reviewer #2 Comment6: The authors tested in vitro stability of selected OEGylated inositol phosphates in enzymatically active human serum or after exposure to 3-phytase (an IP6-degrading enzyme). Importantly (OEG₂)₂-IP4 was active in vivo in rats but metabolism of compounds is often different to humans. Can the authors comment on any further data they may have on whether (OEG₂)₂-IP4 is likely to be effective in humans.

Our response:

We have tested (OEG₂)₂-IP4 on various human materials (e.g., serum, plasma, vascular smooth muscle cells and arterial and valve explants) and observed efficacy and indirect target engagement, with translation to patients as a key driver for performing the studies. While we believe that the provided data have some relevance in humans, we would prefer not to venture a guess on efficacy of (OEG₂)₂-IP4 in humans, given that this can only be properly assessed in clinical trials.

(3) Power calculations and blinding.

Reviewer #2 Comment7: Authors are encouraged to consult the ARRIVE (Animal Research: Reporting of In Vivo Experiments <https://www.nc3rs.org.uk/arrive-guidelines>) guidelines that lists key information to improve reporting of research on animals. The current information is brief and readers would appreciate more information about the study design and reasons for not blinding or carrying out power calculations.

Our response:

We rewrote this paragraph to provide more information on this issue to the readers and the reviewer (p.54, line 1117).

(4) Experimental design and analysis (Fig 4)

Reviewer #2 Comment8 The authors in the rat model of vascular calcification state the numbers in each group as follows: ‘vehicle (n = 10), groups treated with twice daily s.c. injections of 30 mg/kg (OEG₂)-IP4 (n = 11) or IP6 (n = 5), and sham (n = 7). One rat in the vehicle group died suddenly several hours after first vitamin D3 injection without explanation.’ In a study design where the group receiving the treatment/surgery to generate the model is expected to cause some mortality, it is usual to have more to compensate for this than in the sham control. However, the design is the other way round. Please could the authors clarify.

Our response:

Respectfully, there were more animals in the vehicle group (n = 10) than in the sham group (n = 7), so there was no inversion in the design of the experiment.

A new rat was not added in the vehicle group to compensate the observed single death, because of the general reluctance of the veterinarian authorities in Switzerland to perform not absolutely essential animal experiments with reference to the 3R (refinement, reduce, replace). Since an alternative, and less severe vitamin D model was being developed in parallel (vitamin D + warfarin induced rat VC), we did not further pursue this model.

Reviewer #3

Reviewer #3 Comment1: The Design and Synthesis (page 7) of the inhibitors is not clearly described. The account of the chemistry is fragmented, partial, and confusing. It would be far less confusing if the Authors were simply to present the Methods and Data for all twelve compounds, supplemented by NMR spectra to show compound identity and purity. At minimum, the authors should re-write the account to correct the errors (see below) and to explain clearly how and why the compounds were made, giving representative Methods and data for each class of compound.

Our response:

The methods of the two compounds we synthesized ourselves, as well as the methods of the synthesis of the lead compound (OEG₂)₂-IP4 (as provided by the contract research organisation) have been added to the Supplemental Methods (p. 29). Since the other compounds were purchased from Biosynth AG, we cannot provide full details of their synthesis. However, we re-characterized all compounds as well as the intermediates leading to (OEG₂)₂-IP4 by NMR (p. 38 ff.) and MS (p. 48 ff.) to verify identify and purity. We entirely removed two compounds (OEG₂-IP2S3 and OEG₇-IP2S3) and their associated data because the purity assessed by analysis of the NMR spectra was not deemed sufficient.

Specific comments:

Reviewer #3 Comment2: "For initial studies, OEG₁₂-IP5... was synthesized in four steps starting from the protected 1,2,3,4,5-penta-*O*-benzoyl-*myo*-inositol (Supplementary Fig. 1a)." SI Fig 1a does not describe this synthesis, but that of OEG₂-IP5. The synthesis does not start from the penta-*O*-benzoyl compound mentioned. A synthesis of OEG₁₂-IP5 is given in Methods page 36-37, although this does not start from the penta-*O*-benzoyl compound, either.

Our response:

Synthesis of OEG₁₂-IP5 started from 1,3,5-*O*-methylidyne-*myo*-inositol. We have clarified this in the revised manuscript (p. 7, line 134). The synthesis and characterization of OEG₁₂-IP5 is now provided in the Supplemental information (p. 29 ff. Supplemental Methods 3.1.1 and p.41 Appendix NMR Spectra).

Reviewer #3 Comment3: "It was purified... and characterized by [¹H-NMR] Supplementary Fig. 2 and 3), [³¹P NMR] (Supplementary Fig. 4 and 5)..." These figures (pages 68 and 69) give NMR spectra for a completely different compound (OEG₂)₂-IP4, which is commercially available. No methods for the synthesis of (OEG₂)₂-IP4 = INS-3001, the lead compound which features in most of the biology, are given here.

Our response:

Methods of the synthesis and NMR spectra of (OEG₂)₂-IP4 = INS-3001 = lead compound, are provided in the revised version of the Supplemental Methods 3.12. on p. 30 ff. and p. 42, respectively.

Reviewer #3 Comment4: "... as well as [HPLC-UV] (Supplementary Fig. 6)..." This figure (page 70) gives HPLC data (I think!) for the precursor to (OEG₂)₂-IP4 (compound 9), not for OEG₁₂-IP5, nor for (OEG₂)₂-IP4 itself, presumably because these compounds lack good chromophores

for detection by UV, although this is not made clear. ^1H and ^{31}P NMR spectra for all compounds tested would be better, direct, evidence of purity.

Our response:

We rephrased the Figure legend to make it clearer that the benzylated precursors of the synthesized inositol phosphates was used to assess purity, because the final compounds lack good chromophores. We now provide NMR spectra for all compounds in the Supplemental Methods (Supplementary p. 38 ff.).

Reviewer #3 Comment5: "In total, twelve compounds... were produced...and details on their synthesis and characterization can be found in the Supplementary Materials and Methods." The Methods give details for the synthesis of only *two* compounds [(OEG)₁₂-IP5) and (OEG₇)₃-ip3]. These details are well described, but there are no such details for the other ten compounds. The Supplementary gives Schemes (but no Methods or data) for three compounds, plus NMR spectra [wrongly attributed to OEG₁₂-IP5 in the text] for one of these.

Our response:

The response we made to Reviewer #3 Comment 1 clarifies this point.

Corrections:

Reviewer #3 Comment6: Inositol hexakisphosphate, pentakisphosphate and inositol, tetrakisphosphate throughout. Not "hexaphosphate", "pentaphosphate" and "tetrphosphate".

Our response:

This was corrected throughout the entire manuscript and corrections are highlighted in yellow.

Reviewer #3 Comment7: Page 70, legend: "...optically active benzylated precursor..." It is not correct to describe this compound as "optically active" which normally refers to the ability of a compound to rotate the plane of polarized light. All the compounds are either meso-compounds or racemic.

Our response:

By "optically active", we referred to the ability of the benzylated precursor to absorb light. We realize that the wording may be confusing and have deleted these two words and simply state "benzylated precursor". We added a sentence explaining that the synthesized inositol phosphates lack a good chromophore for UV detection and the benzylated precursors can absorb light (Supplementary information 7. Appendix HPLC MS/MS Spectrum).

Reviewer #3 Comment8: Supplementary table 7, page 90: mass spectrometric detection of multiply charged ions ($z = 2+$, $z = 3+$) is the calculation of m/z correct?

Our response:

The calculation was verified and is correct.

Reviewer #3 Comment9: Page 7, Design of compounds: the chelator previously synthesised by this group (ref 42) had the PEG attached to the O2-atom of myo-inositol pentakisphosphate. This is logical because this manner of functionalization gives a meso-compound. The inositol pentakisphosphate derivatives in the present work have OEG attached to the 4/6-O atom, making each compound a racemic mixture, containing two enantiomers with conceivably different biological activities/toxicities. Why did the authors choose this strategy?

Our response:

The bis-4,6 functionalization is the most amenable to a meso bis-OEGylation and the bis-OEG₂ molecule was discovered to be the best compound. Subsequently, the best way to study the structure-activity relationship (i.e., the influence of one vs. two OEGs) was the 4/6-mono-PEG-IP5, despite the ensuing racemic mixture. Activity in the T₅₀ assay of 2-mono-OEG₂-IP5 was comparable to that of the racemic 4/6-mono-OEG₂-IP5 (c350 = 7.9 ± 0.1 vs. 7.1 ± 2.4), which shows that the location of the OEG functionalization does not influence much the compound's activity (Table 1). Given that the compounds target hydroxyapatite surfaces rather than a constrained protein binding pocket, we believe that there is no isomer specificity in the binding.

Table 1 Activity in the T₅₀ assay of selected inositol phosphate derivatives.

Inositol phosphate derivative	Activity in the T ₅₀ assay (c350)	
	μM	mg/L
(±)-2- O -(methoxy-diethyleneglycol)- myo -inositol-1,2,3,5,6-pentakis(phosphate)	8.8 ± 0.1	7.9 ± 0.1
(±)-4- O -(methoxy-diethyleneglycol)- myo -inositol-1,2,3,5,6-pentakis(phosphate)	7.9 ± 2.7	7.1 ± 2.4
4,6-Di- O -(methoxy-diethyleneglycol)- myo -inositol-1,2,3,5-tetrakis(phosphate) (the lead molecule)	4.9 ± 0.8	4.3 ± 0.7

Reviewer #4

Reviewer #4 Comment 1: The authors consider the investigated compound as treatment option. However, clearly the bone mineralization may be affected by the mode of action. The effects of the compound on the bone is not sufficiently investigated, and this limitation not sufficiently discussed, since similar to the adenin model, CKD patients may be the prime patient population for treatment. The bone structure should be documented by μ CT. Is there an effect of the compound on the mechanical properties? Am I missing the effects of the compound on bone in healthy animals – this would seem decisive?

Our response:

In response to the reviewer's comment we investigated the (3D) bone structure of vehicle and (OEG₂)₂-IP4 treated CKD animals by μ CT analysis and compared the results to that of bone samples from control (non-CKD) male Wistar rats of the same age.

We did not include control (non-CKD) groups in the experimental set-up of the current study because of (1) ethical considerations (aiming to limit the number of animals), (2) the primary end-point of the study was to evaluate the effect of (OEG₂)₂-IP4 on the development of vascular calcification, which was of no relevance for control (non-CKD) animals as they do not develop vascular calcification.

Comparison of the results of the 3D bone structure by μ CT of CKD animals of the current study with that of the bone of control (non-CKD) animals, however, enabled us to draw the interesting conclusion that the mineralized trabecular bone structures which are partially lost because of CKD were in fact better preserved in the (OEG₂)₂-IP4 treated animals compared to CKD vehicles (Supplementary Fig. 26 (p. 21) and Supplementary Table 10 (p. 26)).

We were not able to perform analysis of mechanical properties since no appropriate samples for this type of analysis are available (see above, the primary end-point was vascular calcification). We acknowledge that the effect of (OEG₂)₂-IP4 on various aspects of bone in healthy non-CKD animals deserves further investigation. Indeed, such studies will be performed by Inositec Inc. within the scope of GLP toxicology studies in healthy rats and dogs to provide data supporting the initiation of clinical trials. We feel these studies are beyond the scope of the present manuscript, especially considering the feedback of Reviewer 1 regarding the substantial volume of data and complexity already included in the manuscript.

Reviewer #4 Comment 2a: The effects in the animal models should also be characterized better, especially in the uremic model: what are the concentrations of PTH and FGF23?

Our response:

In response to the reviewer's comment we measured concentrations of PTH and FGF-23 in the serum of CKD animals of vehicle and (OEG₂)₂-IP4 groups both at baseline and at sacrifice. As expected both PTH (Rebuttal Letter Fig. 2) and FGF-23 (Rebuttal Letter Fig. 3) were significantly higher at sacrifice compared to baseline. PTH levels did not differ between treatment groups, while FGF-23 was significantly higher in the lowest (5 mg/kg) (OEG₂)₂-IP4 dose groups, compared to vehicle. Since FGF-23 is known as the phosphaturic hormone, the higher serum FGF-23 levels in the (OEG₂)₂-IP4 dose groups are in line with the slightly higher phosphate levels in these lower dose groups. We would propose not to add these results to the manuscript since (i) additional figures and tables have already been included (cf. Reviewer#1 Comment 1 regarding the substantial volume of data already included), (ii) they do not add essential information to the paper and (iii) at the time-point of sacrifice we did not have serum available of all animals, and hence measurements were performed in only 7 (vehicle), 11 (5

mg/kg), 4 (15 mg/kg) and 8 (50 mg/kg) animals. In case the reviewer insists to include these results in the manuscript we would be willing to mention them shortly in the body text of the 'Results' section.

Rebuttal Letter Fig. 2: Serum PTH levels of animals in the adenine-induced VC. The grey region in the background presents the result of non-CKD control animals. Data are expressed as mean \pm s.d.

Rebuttal Letter Fig. 3: Serum FGF-23 levels of animals in the adenine-induced VC model. The result of non-CKD control animals was 0.36 ± 0.06 ng/mL and hence the grey region indicating the baseline is not visible in the figure. Data are expressed as mean \pm s.d. Statistical difference was derived from Kruskal-Wallis test followed by non-parametric Mann-Whitney test with $* p < 0.05$ vs. vehicle.

Reviewer #4 Comment 2b: The beneficial effect on renal function in the Vitamin D model imply, that there may be calcification in the renal system, which causes loss of renal function. What are the effects of treatment on renal calcium content (analogous to aortic calcium content)?

Our response:

We have clarified this point in response to Reviewer #1 Comment 7b.

Reviewer #4 Comment 3a: No clear mechanism of action is described, but some suggestion of a Fetuin-like effect to prevent formation of secCPPs. secCPPs increase intracellular calcium levels, induces oxidative stress and inflammatory responses, as already described mechanisms required for inducing VSMCs calcification. Do the compounds inhibit these changes?

Our response:

We have assessed both H₂O₂ (as proxy for oxidative stress) and TNF- α (as proxy for inflammatory response) in the VSMC culture supernatants subsequent to 48 h –treatment with 50 μ g/mL CPP2 w/ and w/o compounds. However, in both assays we could not detect an increase in neither H₂O₂ nor TNF- α compared to control. Therefore, we were not able to deduce whether the compounds would inhibit these changes^{8,9}. Apart from the two references by Aghagolzadeh and colleagues, to our knowledge no other literature on the effect of CPP2 treatment of VSMC on TNF- α and H₂O₂ levels exist that would allow comparison. We report this in the revised version of the manuscript (p. 11, line 253).

Reviewer #4 Comment 3b: How (OEG₂)₂-IP4 supresses VSMC calcification? Further experiments clarifying at what point (OEG₂)₂-IP4 interferes with the calcification process in vitro would be beneficial.

Our response:

From the herein presented data a clear mechanism of action of how (OEG₂)₂-IP4 supresses VSMC cannot be definitively proven. We hypothesize, however, that (OEG₂)₂-IP4 supresses CPP2 induced VSMC calcification by keeping the particles in suspension and stabilizing the colloid, and thereby preventing either their uptake by VSMCs, and subsequently calcium overload which may cause downstream pro-calcific and pro-inflammatory events. We now discuss this in the manuscript (p. 32, line 614)^{9,10}. Moreover, we now provide further support of the stabilizing effects of the compound on CPPs, by additional *in vitro* data, where (OEG₂)₂-IP4 plus CPP1s do not lead to VSMC calcification (Figure 2g, page 22).

In the case of the CaP-induced VSMC calcification model, the (OEG₂)₂-IP4 most likely acts in multiple ways to prevent calcification, and we discuss this in the revised manuscript (p. 30, line 574):

- i) CPP may form spontaneously in cell culture calcification medium, as evident from dynamic light scattering analysis of the VSMC calcification medium, where at time point 2 days, i.e., when calcification medium was changed, particles of CPP2-size emerged (Supplementary Fig. 5) (Aghagolzadeh, P. et al. 2016)
- ii) Matrix vesicles (MVs), secreted by vascular smooth muscle cells (VSMCs), form the first nidus for mineralization and chemisorption of (OEG₂)₂-IP4 onto such mineralization nidi may result in crystal growth poisoning¹¹.
- iii) at this stage, potential intra-cellular effects cannot be excluded and must be further investigated

Further *in vitro* experiments have been performed and this has been addressed as response to Reviewer #1 Comment 14.

Reviewer #4 Comment 3c: Also, there are then discrepancies in the mechanism of action presented by the authors. On the one hand, based on serum calcification propensity data, the authors suggest that (OEG₂)₂-IP4 inhibits formation of secCPPs and thus, presumably, prevents vascular calcification. However, *in vivo*, the T₅₀ of serum from (OEG₂)₂-IP4 treated animals was not different than from vehicle animals. On the other hand, authors show that (OEG₂)₂-IP4 inhibits VSMCs calcification *in vitro* induced by already formed secCPPs.

Our response:

Respectfully, we would like to mention that there are no discrepancies. The T₅₀ assay describes the propensity of serum for calcification *via* monitoring the time it takes for CPP1s to transition into CPP2s. In the *in vitro* assay, compounds were added right after initiating CPP1 formation, whereas in the T₅₀ assay using the *in vivo* samples, the compound is present in the sample before calcium and phosphate is added to induce CPP1 formation. Moreover, when these samples were collected, calcification was already present, and hence (OEG₂)₂-IP4 was likely strongly associated to calcified deposits in the vasculature (*cf.* also Figure 2j-m) or removed from the circulation by chemisorption to nascent CPPs and their subsequent clearance.

Reviewer #4 Comment 4: Along these lines: The effects on VSMC osteoprotegerin expression is surprising. What are the expression levels of VSMC and calcification markers, such as alpha smooth muscle actin, RUNX2, BMP2, ALPL or TNF-alpha? What is their expression in vascular tissue of the animals? How do authors explain no effects of secCPPs on ALPL activity?

Our response:

The response we made to Reviewer #1 Comment 6 clarifies this point.

Reviewer #4 Comment 5: Fig. 2g. Cytokine levels of TNF α should also be determine in the supernatant of VSMCs. secCPPs-induced VSMC calcification involves increased levels of TNF α and activation TNF α /TNFR1 system in VSMCs.

Our response:

As discussed in the response to Reviewer #4 Comment 3 TNF α levels were assessed but unfortunately not detected. However, to our knowledge there is only a single work from Aghagholzadeh et al., 2016 who have thus far described a significant increase in TNF α levels in the cell culture supernatants of CPP2 treated VSMCs. We have referenced this paper in the main text (p. 11, line 253).

Reviewer #4 Comment 6: The comparative effects and concentration dose responses of the compounds on T50 in conditions without serum should be performed, to determine whether the compounds inhibit directly the precipitation of calcium and phosphate and formation of hydroxyapatite.

Our response:

The response we made to Reviewer #1 Comment 14 clarifies this point.

Reviewer #4 Comment 7: Fig. 4.d-j. Control group without calcification is missing.

Our response:

In order to limit the number of animals used in this experiment, we did not include a control group of rats that did not receive a calcification inducing diet (vitamin D/warfarin). However, during the years, we (i.e., co-authors from University of Antwerp) built up a broad experience with calcification models and exactly know from previous studies which Ca levels might be expected in vessels from control rats. It is important to mention that Ca levels in control vessels are always very low (around 0.2 mg/g tissue¹²) which means that they are almost impossible to show as grey zones on the graphics of Fig. 4 as we did for the bone results (Supplementary Fig. 26, p. 21).

Reviewer #4 Comment 8: Overall, n=3 is a low n number for the in vitro experiments.

Our response:

The *in vitro* cell culture assays presented in Fig. 2 were performed at minimum in 3 independent assays and each assay was performed in 2-well replicates in a 24-well plate. Moreover, some cell culture experiments, e.g., Figure 2e, was conducted to a total of 4 independent assays. Therein, we observed excellent reproducibility and small standard deviations in the results, hence we decided to perform 3 independent assays. In cell culture assays where a 96-well format was used, 3 independent assays using 6-well replicates were done (e.g., Figure 2a (p. 22) and Supplementary Fig. 10a (p. 11)).

Reviewer #4 Comment 9: In figure 1d, why did the authors use 100 μM concentrations of the compounds, while in previous data much lower concentrations were used?

Our response:

A concentration of 100 μM (OEG₂)₂-IP₄ was chosen since no effect was observed at lower concentrations. In this experiment CPPs incubated with 10 μM (OEG₂)₂-IP₄ displayed a morphology in the TEM images that matched those of CPP₂s.

The control group without addition of compounds is missing.

Our response:

The control group can be found in the Supplementary Fig.7 (Supplementary Information, p. 9).

Minor comments

Reviewer #4 Comment 10: The introduction is rather unspecific and superficial when distinguishing between medial, intimal calcification and even valvular calcification. There are some profound differences in these processes. This should be improved. This explains also why the authors used VSMCs and macrophages. This also extends to the discussion

Our response:

Given the multidisciplinary work and complexity of the experimental methodology of this manuscript, it was not possible to go in depth on every topic in the introduction. However, in the process of revising this manuscript in response to the many valuable suggestions made by reviewers, the introduction and discussion has been revised extensively on multiple levels (p. 3 ff. and p. 30 ff., respectively).

Also, there is yet no evidence that reducing calcification is of benefit to mortality.

Our response:

Due to the lack of efficacious pharmacological interventions to reduce and/or prevent vascular calcification, the assessment of causality between reducing VC burden and improving clinical outcome (e.g., mortality) is indeed a challenging endeavor, which we seek to address in the long term. In the present study, however, we show that rats of the vitamin D-induced VC model treated with 2 x 25 mg/kg (OEG₂)₂-IP4 had 100% survival (Fig. 4d, p. 26) and significantly reduced VC compared to vehicle control (Fig. 4e-j). In the vehicle group survival was 63%.

Reviewer #4 Comment 11: Sometimes is written mouse model, p15 last sentence – the methods and figure legends do not clearly state then which animal species and which background was used

Our response:

We have corrected mouse for rat model throughout the entire manuscript. For information on the rat species used, we kindly refer the reviewer to the Methods section 5.2. (p. 50) and 5.5. (p. 54). The rats were not genetically modified.

Reviewer #4 Comment 12: Methods 2.3 is too superficial; calcium and phosphate solutions need to be specified: concentration and what kind of phosphate/calcium was used (or add as described above in 2.1).

Our response:

The text has been amended as suggested by the reviewer (Methods 2.3., p. 41, line 803)

Reviewer #4 Comment 13: Injections in the vitamin D-Warfarin model: “once or twice daily” appears rather random?

Our response:

The rationale herein was to perform a dose-response efficacy study. Therefore, we started with once daily administration of 12.5 mg/kg (OEG₂)₂-IP4 and increased the dosage to twice daily administration of 50 mg/kg (OEG₂)₂-IP4. The text has been rephrased to clarify this (p. 16, line 363)

References

1. Bhattacharjee, S. DLS and zeta potential – What they are and what they are not? *J. Control. Release* **235**, 337–351 (2016).
2. Wang, D., Miller, S., Sima, M., Kopečková, P. & Kopeček, J. Synthesis and evaluation of water-soluble polymeric bone-targeted drug delivery systems. *Bioconjug. Chem.* **14**, 853–859 (2003).
3. Price, P. A., June, H. H., Buckley, J. R. & Williamson, M. K. Osteoprotegerin inhibits artery calcification induced by warfarin and by vitamin D. *Arterioscler. Thromb. Vasc. Biol.* **21**, 1610–1616 (2001).
4. Heiss, A. *et al.* Structural basis of calcification inhibition by α 2-HS glycoprotein/fetuin-A: Formation of colloidal calciprotein particles. *J. Biol. Chem.* **278**, 13333–13341 (2003).
5. Heiss, A. *et al.* Hierarchical role of fetuin-A and acidic serum proteins in the formation and stabilization of calcium phosphate particles. *J. Biol. Chem.* **283**, 14815–14825 (2008).
6. Sato, M. *et al.* Bisphosphonate action: Alendronate localization in rat bone and effects on osteoclast ultrastructure. *J. Clin. Invest.* **88**, 2095–2105 (1991).
7. Zavaczki, E. *et al.* Hydrogen sulfide inhibits the calcification and osteoblastic differentiation of vascular smooth muscle cells. *Kidney Int.* **80**, 731–739 (2011).
8. Aghagolzadeh, P. *et al.* Hydrogen sulfide attenuates calcification of vascular smooth muscle cells via KEAP1/NRF2/NQO1 activation. *Atherosclerosis* **265**, 78–86 (2017).
9. Aghagolzadeh, P. *et al.* Calcification of vascular smooth muscle cells is induced by secondary calciprotein particles and enhanced by tumor necrosis factor- α . *Atherosclerosis* **251**, 404–414 (2016).
10. Köppert, S. *et al.* Cellular Clearance and Biological Activity of Calciprotein Particles Depend on Their Maturation State and Crystallinity. *Front. Immunol.* **9**, 1–17 (2018).
11. Kapustin, A. N. *et al.* Vascular smooth muscle cell calcification is mediated by regulated exosome secretion. *Circ. Res.* **116**, 1312–1323 (2015).
12. Persy, V. *et al.* High-resolution X-ray microtomography is a sensitive method to detect vascular calcification in living rats with chronic renal failure. *Arterioscler. Thromb. Vasc. Biol.* **26**, 2110–2116 (2006).

REVIEWERS' COMMENTS:

Reviewer #1 (Remarks to the Author):

The revised version of the manuscript fully addresses all concerns raised by this reviewer. Further supporting data has been added to the already impressive previous version.

p4 li 76-78, As an option, I suggest a slight change in phrasing a newly added sentence in the introduction to better reflect current knowledge. "Endogenous CPPs represent a highly heterogeneous mixture of particles with characteristics mostly intermediate to synthetic CPP1 and CPP2 (ref 27)." should be changed to "Native CPP in fresh plasma are smaller in size and lower in density than synthetic CPP1 and CPP2 (ref 29), and may in fact represent mineral laden fetuin-A monomer termed calciprotein monomer CPM in a biophysical study of fetuin-A as a mineral carrier. (Heiss et al. Biophysical J. 2010, doi: 10.1016/j.bpj.2010.10.030

signed W Jahnen-Dechent

Reviewer #2 (Remarks to the Author):

The authors have endeavoured to respond to all the referees questions in detail and to make changes where appropriate in the manuscript.

Reviewer #3 (Remarks to the Author):

The authors have now substantially addressed my chemistry-related criticisms and answered my queries.

They have corrected errors, reorganised parts of the ms and added additional characterization data.

Two compounds have been removed from the ms because their purity was insufficient as judged by NMR.

Affected compounds have been re-numbered accordingly.

I can only comment on the chemistry-related parts of the ms, but I now consider the work to be suitable for publication from a chemistry point of view.

Reviewer #4 (Remarks to the Author):

The authors addressed the noted questions, and present a thorough and insightful manuscript. Especially the μ CT measurement add valuable information. The authors present a plausible hypothesis (which remains to be definitively proven) about the mechanisms of action of the (OEG2)2-IP4 compound.

Please add in the discussion a concise statement, why (OEG2)2-IP4 compound could delay T50 in isolated experiments but not in rat serum.

REVIEWERS' COMMENTS:

Reviewer #1 (Remarks to the Author):

p4 li 76-78, As an option, I suggest a slight change in phrasing a newly added sentence in the introduction to better reflect current knowledge. "Endogenous CPPs represent a highly heterogeneous mixture of particles with characteristics mostly intermediate to synthetic CPP1 and CPP2 (ref 27)." should be changed to "Native CPP in fresh plasma are smaller in size and lower in density than synthetic CPP1 and CPP2 (ref 29), and may in fact represent mineral laden fetuin-A monomer termed calciprotein monomer CPM in a biophysical study of fetuin-A as a mineral carrier. (Heiss et al. Biophysical J. 2010, doi: 10.1016/j.bpj.2010.10.030

We thank the reviewer for his suggestion, and adopted the sentence partially. It now reads: "Native CPPs in fresh plasma are smaller in size and lower in density than synthetic CPP1 and CPP2²⁹"

Reviewer #4 (Remarks to the Author):

Please add in the discussion a concise statement, why (OEG₂)₂-IP₄ compound could delay T₅₀ in isolated experiments but not in rat serum.

This was added to the discussion on page 32, line 623: "No delay in T₅₀ was observed when serum samples were tested from the adenine-induced VC rat model, as (OEG₂)₂-IP₄ was likely to be already eliminated or bound to existing calcified deposits at the time of sampling."